# Comparison of ocean heat content estimated using two eddy-resolving hindcast simulations based on OFES1 and OFES2

Fanglou Liao[1,2], Xiao Hua Wang[2*], and Zhiqiang Liu[1,3*]

[1]Department of Ocean Science and Engineering, Southern University of Science and Technology, Shenzhen, 518055, China

[2]The Sino-Australian Research Consortium for Coastal Management, School of Science, The University of New South Wales, Canberra, 2610, Australia

[3]Southern Marine Science and Engineering Guangdong Laboratory (Guangzhou), Guangzhou, 511458, China

*Correspondence to*: Zhiqiang Liu (liuzq@sustech.edu.cn) or Xiao Hua Wang (x.h.wang@unsw.edu.au)

**Abstract.** In this study, we have compared the ocean heat content (OHC), estimated using two eddy-resolving hindcast simulations based on Ocean General Circulation Model for the Earth Simulator Version 1 (OFES1) and Version 2 (OFES2). Results from a global objective analysis of subsurface temperature (EN4) were taken as a reference. Both EN4 and OFES1 suggest that OHC has increased in most regions of the top 2000 m during 1960–2016, which is mainly associated with the deepening of neutral density surfaces and variations along the neutral density surfaces of regional importance. Upon comparing the results obtained from the two OFES hindcasts, we found substantial differences in the temporal and spatial distributions of the OHC, especially in the Atlantic Ocean. A basin-wide heat budget analysis showed that there was less surface heating for the major basins in the OFES2. The horizontal heat advection was mostly similar, however, the OFES2 had a significantly stronger meridional heat advection associated with the Indonesian Throughflow (ITF) above 300 m. Additionally, large discrepancies in the vertical heat advection were also evinced when the two OFES results were compared, especially at a depth of 300 m in the Indian Ocean. We inferred that there are large discrepancies in the vertical heat diffusion (cannot be directly evaluated in this study due to data unavailability), which, along with the different magnitudes of sea surface heat flux and vertical heat advection, were the major factors responsible for the examined differences in OHC. This work suggests that OFES1 provides a reasonable multi-decadal estimate of global and basin-integrated warming trends above 700 m, except for the top 300 m for the Pacific Ocean and between 300–700 m for the Indian Ocean. Although the estimates of the global OHC during 1960–2016 are consistent with observations between 700–2000 m, caution is warranted while examining the basin-wide multi-decadal OHC variations using OFES1. The seemingly suboptimal OHC estimate based on OFES2, suggests that any conclusions on long-term climate variations derived from OFES2 might suffer from large drifts, necessitating audits.

## 1 Introduction

The global oceans store more than 90% of extra heat that has been added to the Earth since the 1950s, generating a significant OHC increase (Levitus et al., 2012; IPCC 2013). Therefore, OHC forms an important indicator of climate change, and it helps estimate the Earth's energy imbalance (Palmer et al., 2011; Von Schuckmann et al., 2016). Although natural factors such as El Niño–Southern Oscillation (ENSO) and volcanic eruptions can modulate the OHC

(Balmaseda et al., 2013; Church et al., 2005), the recent warming trend has been largely induced by the accumulation
of greenhouse gas in the atmosphere (Abraham et al., 2013; Gleckler et al., 2012; Pierce et al., 2006).
The OHC increase, being a major concern for both oceanography and climate communities, has attracted a great deal
of attention. Although direct observational records represent the most reliable data for determining the oceanic thermal
state, the available observations are not dense enough in both the temporal and spatial domains, especially for the deep
and abyssal oceans. The number of observations has greatly improved since the launch of a global array of profiling
floats, the Argo, in the 2000s. However, the spatial resolution of the Argo program (i.e., approximately 300 km) is not
high enough to capture mesoscale structures (Sasaki et al., 2020, hereafter **S2020**). There are several approaches for
filling the temporal and spatial gaps in global temperature measurements, which can be used to produce gridded
temperature products for estimating the OHC. Typical approaches include an objective analysis (Good et al., 2013) of
observational data and an ensemble optimal interpolation with a dynamic ensemble (EnOI-DE (Cheng and Zhu, 2016).
In addition, ocean general circulation models (OGCMs) provide the temperature fields by solving primitive equations
of fluid motion and state. When constrained by observations, a numerical ocean modelling becomes the ocean
reanalysis, which geneally lacks dynamical-consistence (the resulting fields satisfy the underlying fluid dynamics and
thermodynamics equations), unless the adjoint method was adopted to use information contained in observations.
Although ocean reanalysis has been widely constructed, unconstrained OGCMs are still an important tool for climate
prediction, for instance, the Coupled Model Intercomparison Project (CMIP). How multi-scale dynamical processes
are represented in these unconstrained models and their implementation of external forcing significantly impact their
OHC estimates.
The Ocean General Circulation Model for the Earth Simulator (OFES (Masumoto et al., 2004; Sasaki et al., 2004)),
developed by the Japan Agency for Marine-Earth Science and Technology (JAMSTEC) and other institutes, is a well-
known eddy-resolving OGCM, and the hindcast simulation of the OFES Version 1 (OFES1) has been widely used
(Chen et al., 2013; Dong et al., 2011; Du et al., 2005; Wang et al., 2013). The hindcast simulation based on the OFES
Version 2 (OFES2) has now been released with certain improvements over the OFES1 (**S2020**). For example, in a
comparsion to the OFES1, the authors found a smaller bias in the global sea surface temperature (SST), sea surface
salinity (SSS), and the water-mass properties of the Indonesian and Arabian seas in the OFES2. To our knowledge,
however, a comparison of the multi-decadal OHC at a basin or global scale between OFES1 and OFES2 is lacking.
As this high-resolution quasi-global hindcast simulation is expected to be widely used in oceanography and climate
communities for examining the state of the ocean in the near future, it is necessary to compare the OHC estimated
using the two OFES as an indicator of the potential improvements in OFES2 over OFES1. Such a study is also
expected to provide insights on the adaptability of the two simulations for OHC-related studies. The finding that
subsurface oceanic fields could be notably different when estimated based on the results of two OFES runs with
different atmospheric forcing, despite their similar results in the near-surface region (Kutsuwada et al., 2019), forms
an added motivation to conduct the envisioned study.
The aim of this study is twofold: (1) estimate the OHC in the global ocean and in each major basin using OFES1 and
OFES2, with a primary focus on differences between the two hindcasts; and (2) understand the causes of the
differences between the two hindcasts. To this end, we used the potential temperature $\theta$ to calculate and compared the
OHC from 1960 to 2016 for both the global ocean and the major basins, i.e., the Pacific Ocean, the Atlantic Ocean,
and the Indian Ocean between 64° S and 64° N.
In Section 2, we provide a brief description of the data and methods used in this study. In Section 3, we describe and
discuss the differences in OHC in both the temporal and spatial domains. A tentative analysis of the possible causes
of these differences was also conducted. Section 4 summarizes the principal points and the possible extensions
involving factors that were not examined here due to data unavailability, although such factors could be important.
Accordingly, we have added the future scope of this study to improve the associated work.
**2 Data and Methods**
**2.1 Data**
The potential temperature $\theta$ from both OFES1 and OFES2 were used to calculate the global and basin OHCs. This
allowed us to compare the OHC estimated from OFES1 and OFES2, along with the estimates from the observation-
based EN4. Although results from EN4 cannot be considered to represent the actual oceanic state, it has been widely
used in OHC-related studies (Allison et al., 2019; Carton et al., 2019; Häkkinen et al., 2016; Trenberth et al., 2016;
Wang et al., 2018). A brief description of the three datasets is given below; readers are referred to Sasaki et al. (2004),
Sasaki et al. (2020), and Good et al. (2013) for a more detailed description.
The OFES1 has a horizontal spatial resolution of 0.1° with 54 vertical levels and a maximum depth of 6065 m (Sasaki
et al., 2004). Such a high lateral resolution enables it to resolve mesoscale processes. Following a 50-year
climatological simulation, the hindcast simulation of the OFES1 was integrated forward, with the publicly available
data from 1950 to 2017. The multi-decadal integration made it possible to analyze oceanic fields at temporal scales
from intra-seasonal to multi-decadal. Unlike most other datasets used for the estimation of the OHC, OFES1 is
unconstrained by any observations. Therefore, it can be used to demonstrate the adaptability of high-resolution
numerical modeling without data assimilation in climate studies.
OFES2 has the same horizontal spatial resolution of 0.1°. Vertically, there are 105 levels with a maximum depth of
7500 m. OFES1 uses National Centers for Environmental Prediction (NCEP) reanalysis (2.5° × 2.5° (Kalnay et al.,
1996)) for atmospheric forcing on an everyday basis, whereas OFES2 obtains atmospheric forcing from the JRA55-
do Version 08 (55 km × 55 km (Tsujino et al., 2018)) with a temporal resolution of 3 hours. Both the temporal and
spatial resolutions of atmospheric forcing have increased significantly in the OFES2. The OFES2 also incorporates
river runoff and sea-ice models, but polar areas are not included.
In the horizontal direction, both OFES1 and OFES2 use a biharmonic mixing scheme to suppress the computational
noise (**S2020**). The horizontal diffusivity coefficient is equal to $-9{\times}10^9$ m$^4$/s at the equator (**S2020**) and varies
proportionally with the cube of the cosine of the latitude (personal communication with Hide Sasaki). The OFES2
uses a mixed-layer vertical mixing scheme (Noh and Kim 1999) with parametrization of tidal energy dissipation (Jayne
and St. Laurent 2001; St. Laurent et al., 2002), whereas OFES1 uses the K-profile parameterization (KPP) scheme
(Large et al., 1994). Taking the temperature and salinity of January 1, 1958 from OFES1 as the initial conditions,
OFES2 was integrated forward, with the publicaly available data from 1958 to 2016. To reduce the computation time,
we subsampled the OFES1 and OFES2 data at every five grid points in the horizontal direction.
To evaluate the OHC from the two OFES data, we used EN4 from the UK Meteorological Office Hadley Centre as
a reference. Note that we used the EN4.2.1 as the EN4 version, with bias-corrected following Levitus et al. (2009).
The EN4 data can be considered as objective analysis data that is based on observations (Good et al., 2013), with a
horizontal resolution of 1° and 42 vertical levels down to 5350 m. The EN4 assimilates data mostly from the World
Ocean Database (WOD) and the Coriolis dataset for ReAnalysis (CORA). Preprocessing and quality checks were
conducted before the observational data were used to construct this objective analysis product.
Although we used the results from EN4 as a reference for evaluating the performance of OFES in simulating the 57-
year thermal state of the ocean, EN4 cannot be considered to represent the actual ocean state. The main reason is that
the measurements used to construct the EN4 datasets are sparse and inhomogeneous in both temporal and spatial
domains, and are insufficient to resolve mesoscale or even sub–mesoscale motions. There are more observations in
the northern hemisphere compared to the southern hemisphere, and there is also a seasonal bias in the observational
data density (Abraham et al. 2013; Smith et al. 2015). A larger density of data was generated only after the World
Ocean Circulation Experiment (WOCE) was conducted in the 1990s and following the launch of the Argo profiling
floats in the 2000s. Table 1 summarizes the three datasets.
**Table 1.** A summary of the OFES1, OFES2 and EN4. The symbol / means "not applicable".

|  | OFES1 | OFES2 | EN4 |
|---|---|---|---|
| Model | MOM3 | MOM3 | / |
| Horizontal coverage | 75° S – 75° N | 76° S – 76° N | 83° S – 89° N |
| Horizontal grids | 3600 × 1500 | 3600 × 1520 | 360 × 173 |
| Vertical levels | 54 | 105 | 42 |
| Maximum depth | 6065 m | 7500 m | 5350 m |
| Atmospheric forcing | NCEP | JRA55–do Ver.08 | / |
| Data assimilated | / | / | WOD, CORA |
| Time span | 1950 – 2017 | 1958 – 2016 | 1900 – 2021 |


We considered water from the sea surface to approximately 2000 m, and divided it into three layers: upper (0–300
m), middle (300–700 m), and lower (700–2000 m). The ocean above 2000 m is often divided into two layers, 0–700
m and 700–2000 m (or even one: 0–2000 m) (Allison et al., 2019; Häkkinen et al., 2015; Häkkinen et al., 2016; Levitus
et al., 2012; Zanna et al., 2019). However, our analysis shows that it is necessary to divide it into three layers to reach
the objective of this study. Similar vertical division can also be seen in Liang et al. (2021).
The reasons for ignoring water below 2000 m were mainly fourfold. First, the simulated behavior of the deep and
abyssal oceans depends on the spin-up of the numerical simulation, which is mostly incomplete (Wunsch 2011), at
least in the first decade. Second, the observational data used in EN4 are largely confined to the top 2000 m, and some
available measurements do not even go down to this depth (personal communication with the EN4 UK Meteorological
Office Hadley Centre). The number of data is significantly lesser in the deep and abyssal oceans. Third, the EN4 data
that we used here was bias-corrected following  Levitus et al. (2009), in which only the ocean above 700 m was
considered. For instance, the Expendable Bathythermograph (XBT) profiles below 700 m were corrected using the
correction values provided for 700 m (personal communication from the UK Meteorology Office Hadley Centre).
Finally, the maximum depth of OFES2 and EN4 differs by more than 2000 m. It was felt that the full-depth OHC,
estimated using the three datasets, is not highly comparable. However, this does not imply that we can ignore the
contribution of the deep ocean; it can play an essential role in regulating the global-ocean thermal state (Desbruyères
et al. 2016; Desbruyères et al. 2017; Palmer et al. 2011). It is expected that a significantly better understanding of the
deep and abyssal ocean states will be gained with the implementation of the Deep Argo program, which is partially
validated by Johnson et al. (2019).
**2.2 Methods**
We compared the three datasets for the period 1960–2016. In this paper, the OHC represent the OHC anomalies
relative to the OHC estimates of 1960. At each grid point, the OHC is expressed as follows:

$$\text{OHC} = \rho \delta v C_p (\theta - \theta_{1960}) = \rho \delta v C_p \Delta \theta, \tag{1}$$

where $\rho$ is the seawater density (kg/m$^3$), $\delta v$ is the grid volume (m$^3$), $C_p$ is the specific heat of seawater at constant
pressure (J/kg/°C), $\theta$ is the yearly potential temperature (°C), and $\theta_{1960}$ is the average potential temperature during
1960. The total OHC in the upper ocean layer (above 300 m) is the integral of Eq. (1) from 0 to 300 m. Similar
procedures were applied to the other two layers (300–700 m and 700–2000 m). A value of $4.1 \times 10^6$ J/m$^3$/°C was used
for the product of $\rho$ and $C_p$ (Palmer et al., 2011).
OHCs of both global and individual basins were calculated for comparison. Fig. 1 shows the domains of the Pacific,
Atlantic, and Indian Oceans between 64° S and 64° N, including their respective marginal seas. Our definition of the
marginal seas of each major basin may be inconsistent with those of other studies. The major water passages
connecting the different basins are denoted by red lines in Fig. 1a. Fig. 1b is the schematic of primary processes that
determine the OHC of an ocean basin.

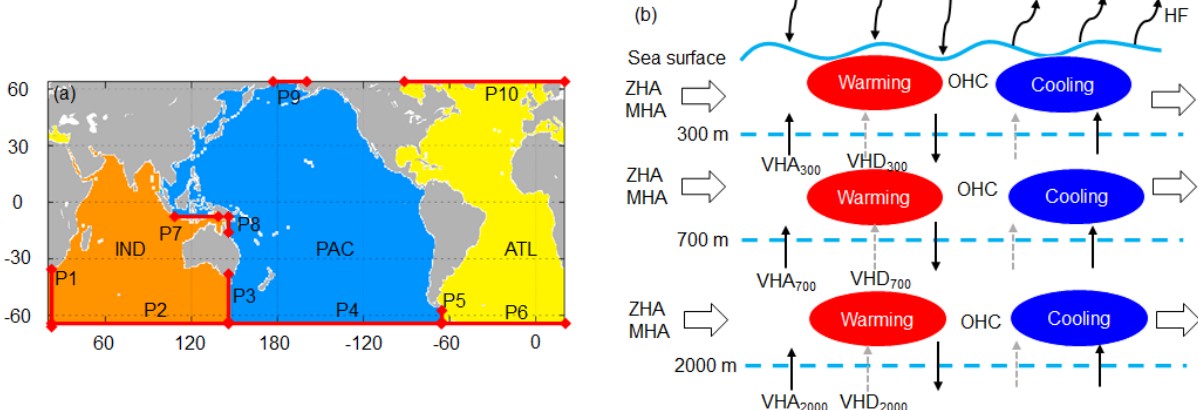


**Figure 1.** (**Left**) Domains of the major basins between 64° S and 64° N and (**right**) a schematic of the primary
processes controlling the thermal state of an ocean. (**a**) The PAC stands for the Pacific Ocean, the ATL for the Atlantic
Ocean and the IND for the Indian Ocean. The basin domain is extracted using the gcmfaces package (Forget et al.,
2015) and then interpolated to the corresponding grid of each product. Grey indicates the land. The red solid lines
with diamond arrow stand for the water passages connecting different basins. We label it with the capital letter P
(abbreviation for passage) and a serial number. The horizontal and vertical axis are longitude and latitude, respectively.
(**b**) We use a light blue solid curve to represent the free sea surface and three dashed lines to indicate the 300 m, 700
m and 2000 m depth. The curve arrow represents the net heat flux (HF) through the ocean surface. The black hollow
arrow shows the zonal (ZHA) or meridional (MHA) heat advection. The black thin arrow represents the vertical heat
advection (VHA) and the grey dash arrow stands for the vertical heat diffusion (VHD). The red ellipse illustrates
warming water and the blue ellipse cooling water. P1: (20° E, 64° S – 34.5° S); P2: (20° E – 146.5° E, 64° S); P3:
(147° E, 64° S – 36.5° S); P4: (147° E – 65.5° W, 64° S); P5: (67° W, 64° S –55° S); P6: (65° W – 19.5° E, 64° S);
P7: (118.5° E – 138.5° E, 8.5° S); P8: (142° E, 12.5° S – 8° S); P9: (172.5° W – 166.5° W, 64° N); P10: (88° W –
19.5° E, 64° N).

In addition, $\Delta\theta$ at a fixed depth is decomposed into a heave (HV) component (the second term of Eq. (2)) and a
spiceness (SP) component (the third term of Eq. (2)) (Bindoff and McDougall, 1994). HV-related warming or cooling
is manifested as a vertical displacement of the neutral density surfaces (a continuous analog of discretely referenced
potential density surfaces (Jackett and McDougall, 1997)). In general, both the dynamic changes and the change in
the renewal rates of water-masses can induce vertical displacement, generating HV-related warming or cooling
(Bindoff and McDougall, 1994). SP represents warming or cooling as a result of density compensation in $\theta$ and salinity
($S$) along the neutral density surfaces. Decomposition of $\Delta\theta$ helps to better understand the contributions of different
water-masses to generating OHC. The formula for decomposing the potential temperature is given as follows:
$$d\theta/dt\,|_z = - \overbrace{dz/dt|_n\, d\theta/dz}^{HV} + \overbrace{d\theta/dt|_n}^{SP} \,, \qquad (2)$$
where $t$ is the time (year), $z$ is the depth (m), and $|_n$ means along the neutral density surface.
The program developed by Jackett and McDougall (1997) was used to calculate the neutral densities, HV, and SP.
This code is based on the United Nations Educational, Scientific and Cultural Organization (UNESCO), 1983 for the
computation       of       fundamental       properties       of       seawater       (http://www.teos–
10.org/preteos10_software/neutral_density.html). We used its MATLAB version for our calculations. The main inputs
for this program were $\theta$ and $S$. The code limits the latitude to be between 80° S and 64° N, but we further confined
our investigation domain to 64° from the equator, which avoids comparisons in sea-ice-impacted areas, given that
only OFES2 includes a sea-ice model.
To analyze the origin of the differences in OHC from thermodynamic and dynamic perspectives, we calculated the
surface heat flux (HF), zonal heat advection (ZHA), meridional heat advection (MHA), and vertical heat advection
(VHA). Owing to a temporary suspension of the OFES2 data by the JAMSTEC, we could not access the vertical
diffusivity data of OFES2 while preparing this manuscript. Note that OFES1 does not provide such data. This
prevented us from directly comparing the estimates of vertical heat diffusion (VHD) based on OFES1 and OFES2.
Alternatively, we calculated the residual of the total OHC and all the other heat inputs (HF, ZHA, MHA, and VHA),
and used the results as a proxy for VHD. As the horizontal heat diffusion was found to be significantly weaker than
that of ZHA and MHA (not shown), we did not include it in the analysis. A schematic of the primary process is shown
in Fig. 1b. Note that the linear trend in the following sections was calculated using multiple linear regression using
least squares at 95% confidence level.
**3 Results**
The principal objective of this study is to compare the results from OFES1 and OFES2, considering EN4 as an
observation-based reference. We attempted to evaluate if there is any significant difference between the results
obtained from OFES2 and those from one or both of the other two datasets, and if any such difference represents a
real phenomenon that is not present in the other two widely used datasets or it is an unwanted property of the newly
released OFES2 simulation. In this section, we compare the three sets of results for the global ocean, along with
individual cases of the Pacific, Atlantic, and Indian Oceans.

**3.1 Temporal evolution of the OHC, HV, and SP from 1960 to 2016**

**3.1.1 Time series of OHC, HV, and SP**

Figures 2–4 illustrate the time series of the total OHC, and its HV and SP components for the upper (0–300 m), middle
(300–700 m), and lower (700–2000 m) ocean layer, respectively. Note that OHC, HV, and SP were calculated as an
anomaly relative to the estimates in 1960, and was converted to an equivalent HF applying over the entire surface area
of the Earth.

*Upper layer*
For the global ocean between 0 and 300 m, all three data indicate cooling from approximately 1963 to 1966 (Fig. 2a),
which was caused by the volcanic eruption of Mount Agung (Balmaseda et al., 2013). A similar trend of cooling
during this period is also reported for the upper 700 m (Domingues et al., 2008; Allison et al., 2019) and for both 0–
700 m and 0–3000 m depth (Achutarao et al., 2007). This short, however, sharp cooling period significantly impacted
the Pacific Ocean (Fig. 2b). Marked reductions in the OHC associated with strong volcanic eruptions of El Chichón
in 1982 (a strong El Niño also emerged in 1982–83) and Pinatubo in 1991 were also consistently captured by all three
data.

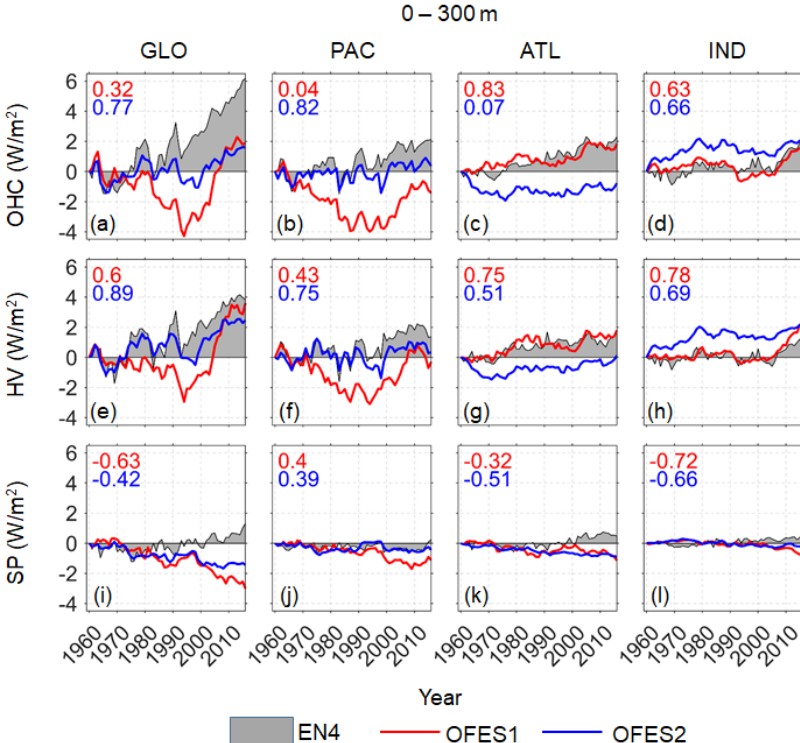


**Figure 2.** Time series of the global and basin-wide OHC (**top**), HV (**middle**) and SP (**bottom**) between 0–300 m
based on the three datasets. The OHC, HV and SP here are converted to the accumulative heating in W/m² applied
over the entire surface of Earth. Grey shadow: EN4; red solid line: OFES1; blue solid line: OFES2. Numbers on the
left top corners are the correlation coefficients between the OFES1 (red) or OFES2 (blue) and EN4. The OHC hereafter
is directly calculated from the potential temperature, rather than the sum of the HV and SP.

Both EN4 and OFES2, but not OFES1, showed a slowdown in warming in the Pacific Ocean during the 2000s (Fig.
2b). This slowdown of warming in the Pacific corresponds to a sharp warming trend in the upper layer of the Indian
Ocean (Fig. 2d), seen in all the three datasets. This relationship between the Pacific and Indian Oceans could be a
consequence of intensified Indonesian Throughflow (ITF), which increased heat transport from the Pacific to the
Indian Oceans (Lee et al. 2015; Zhang et al. 2018). Note that these two studies considered the top 700 m. However,
the sudden warming of the Indian Ocean was largely confined to the top 300 m, which is indicated by OFES1 and
OFES2 (Fig. 3d). The EN4 showed a clear acceleration of warming trend above 300 m in the global ocean around
2003, which was probably an artifact caused by the transition of the ocean observation network from a ship-based
system to Argo floats (Cheng and Zhu, 2014), although these authors mainly used subsurface temperature data from
the World Ocean Database 2009 (WOD09). Interestingly, a dramatic shift can also be seen in OFES1 (Fig. 2a),
although that OFES1 is not directly constrained by observations. A major difference in this jump between EN4 and
OFES1 is its close association with SP in EN4 (Fig. 2i) compared to HV in OFES1 (Fig. 2e). This spiciness warming
around 2003, derived from EN4, complements the work of Cheng and Zhu (2014).
However, several significant differences were observed between the three datasets. Results from EN4 indicated that
the temporal evolution of the warming was approximately linear since ~1970 (Fig. 2a), which was modulated by the
abovementioned climate signals. The OFES1, however, showed that the cooling period persisted almost until the early
1990s, while a stronger linear warming trend appeared afterward (Fig. 2a). This was more than 20 years later than that
indicated by the EN4. In the OFES2, the approximately linear warming trend appeared even later ( ~2000), the
magnitude of which was approximately the weakest among the three datasets.
Compared to OFES1, the temporal profile of the global upper ocean obtained using OFES2 generally agreed better
with that indicated by EN4 (Fig. 2a), which, to some extent, is consistent with the smaller SST bias estimated from
the OFES2 than that from the OFES1 when compared to the World Ocean Atlas 2013 (WOA13) (**S2020**). However,
the difference between OFES2 and EN4 in magnitude became larger after 1980. This was mainly due to the SP
component (Fig. 2i), with both OFES1 and OFES2 indicating a clear SP cooling episode. This might imply some
discrepancies in the salinity information of these three datasets. In contrast, there was a good agreement between the
HV values of EN4 and OFES2 (Fig. 2e).
Clear differences can also be seen for each basin. The OFES1 differed significantly from the other two in the Pacific
Ocean during 1970–1990, with the other two being similar to each other with respect to both HV and SP. In the
Atlantic Ocean, however, the OFES1 agreed quite well with the EN4 in the HV. The two OFES datasets had similar
spiciness in the Atlantic Ocean, but both disagreed with the spiciness of EN4. The HV, estimated using OFES2,
showed poor agreement with both EN4 and OFES1 in the 1960s (Fig. 2g). In the Indian Ocean, OFES1 was much
closer to EN4 than OFES2. The notable deviations of the OFES2 relative to others were mainly generated from the
uniquely strong warming trend in the OFES2 Indian Ocean before ~1980 (Fig. 2d).
A potential issue of the OFES2 is the spin-up, although it was initiated from the calculated temperature and salinity
fields from OFES1. Without any prior knowledge of the timing of complete spun-up, here we have shown and
compared the simulated results from 1960, excluding the first two years (1958–1959). It seems that the results obtained
using OFES2 have a better agreement with EN4 since the 1980s for both Atlantic and Indian Oceans (Fig. 2c, d),
which is likely to be related to the improvement in spun-up with time. However, in the Pacific Ocean, the OFES2 was
quite similar to EN4 before 1990, especially its HV component. This, to some extent, might weaken the spin-up
argument.

*Middle layer*
In the middle ocean layer (300–700 m) (Fig. 3), there were remarkable differences in the OHC and its HV and SP
components between the OFES2 and the other two datasets, which is most noticeable in the Atlantic Ocean, and lesser
for the Pacific Ocean; the difference was minor for the Indian Ocean. The OFES2 showed a moderate Pacific cooling
for almost the entire 57–year period and a strong Atlantic cooling trend until ~2000, with a subsequent hiatus in the
Atlantic Ocean. The OFES2 indicated that there was a minor cooling in the Indian Ocean during the 1960–70s. In
OFES2, these uniquely cooling trends were mainly associated with HV because its spiciness was generally more
positive than as indicated by the other two datasets.

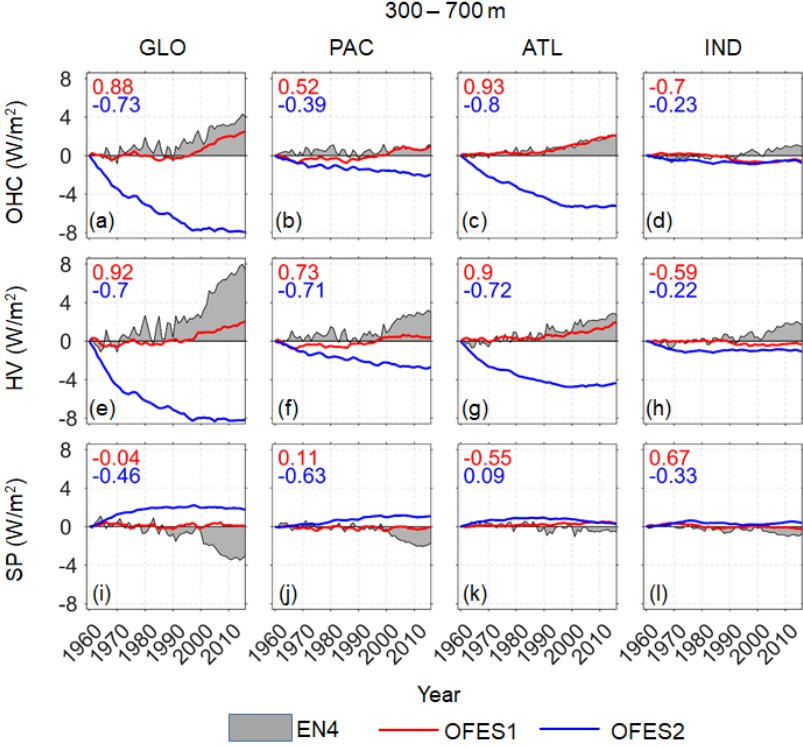


**Figure 3.** As for Fig.2 but for the middle layer (300–700 m).

In contrast, both EN4 and OFES1 indicated that the middle layer was relatively stable before the early 1990s (Fig.
3a). Afterwards, EN4 and OFES1 both showed global ocean and Atlantic Ocean warming (Fig. 3a, c), mostly due to
an increase in the HV (Fig. 3e, g). Despite such good agreement between EN4 and OFES1, there were notable
differences in their HV and SP components. Compared to the OFES1, there was a stronger positive HV in the EN4
(Fig. 3e–h) and a stronger negative SP in the EN4, particularly after 2000 (Fig. 3i, j). A possible reason for this finding
may be that  many more observations have become available since the WOCE was conducted in the late 1990s and
from the Argo since the beginning of the 2000s. This might have led to a systematic trend in the observation-based
dataset EN4. Unlike EN4 and OFES2, the SP variations in OFES1 were almost invisible for almost all the basins. In
addition, the aforementioned significant warming acceleration from the early 2000s to the 2010s in the Indian Ocean
(Fig. 2d) can still be seen in the EN4 (Fig. 3d), however, this was almost invisible in the two OFES datasets.
One major cause of the profound differences between OFES2 and the other two datasets may be the spin-up issue.
Indeed, even after 2000, clear differences can be observed in the global ocean. This is expected because the middle
layer takes more time to be completely spun compared to the upper layer. Hence, special caution is required while
investigating the multi-decadal variations or even decadal variations in the recent two decades based on OFES2.

*Lower layer*
In the lower oceanic layer (700–2000 m) (Fig. 4), the OFES2 was again an outlier among the three datasets. It showed
that the Atlantic and the Indian Oceans experienced cooling from 1960 to the end of the 1990s (Fig. 4c, d), followed
by a slight warming episode. In the Pacific Ocean, however, OFES2 showed cooling over the entire 57-year period
(Fig. 4b). The better agreement between the results from OFES2 and EN4 since the end of the 1990s might be related
to the spin-up issue of the OFES2, at least to some extent. However, the agreement between EN4 and OFES2 was
even better than that in the middle layer (300–700 m), particularly in the Atlantic Ocean. This might weaken the spin-
up argument because it is expected that the middle layer can be more easily spun-up than the lower layer.
The variations in OHC determined using OFES1 and EN4 were similar for the global ocean, however, this could be
associated with the cancelation of the substantial differences in the Pacific and Atlantic Oceans (Fig. 4b, c), and in the
HV and SP (Fig. 4e–l). More specifically, there was a larger increase of OHC in the Pacific Ocean, when estimated
using OFES1 than from EN4, however, the latter showed a larger increase of OHC in the Atlantic Ocean. From the
perspective of potential temperature decomposition, EN4 generally showed a stronger increase in HV than OFES1 in
the Atlantic and Indian Oceans (Fig. 4g, h), however, a stronger negative or a weaker positive increase of SP is also
evinced (Fig. 4i–l).

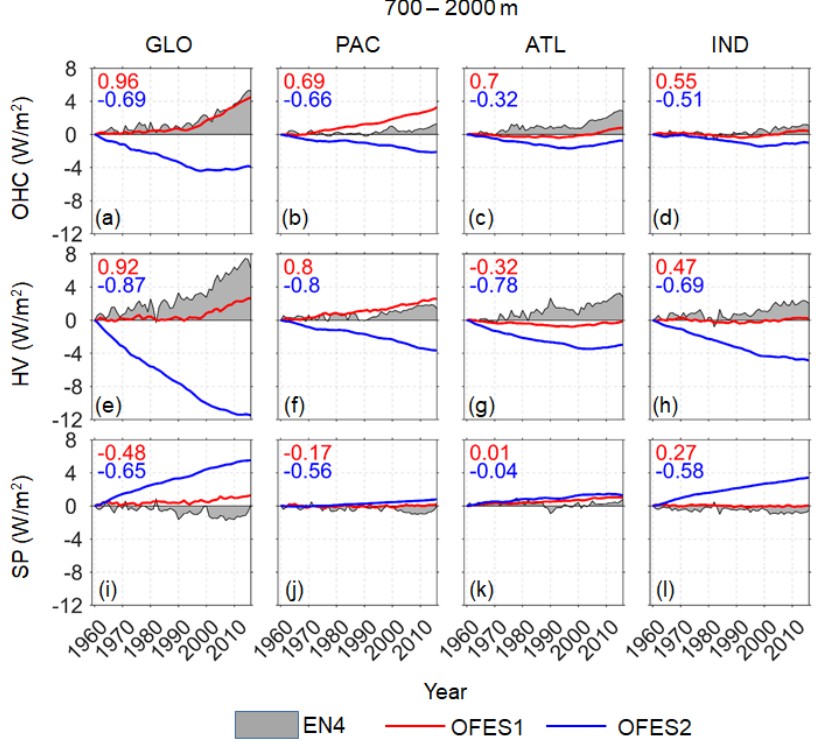


**Figure 4.** As for Fig.2 but for the lower layer (700–2000 m).

**3.1.2 Temporal evolution of the OHC, HV, and SP trend**
Figures 2–4 show the similarities and differences between the three datasets with respect to the time series of OHC,
HV, and SP for the period 1960–2016. In this section, we calculate the linear trend in OHC, HV, and SP over a rolling
window of 10 years for the three datasets following Smith et al. (2015), and the results for the three layers are shown
in Figures 5–7, respectively. Such evaluation has helped us to quantitatively compare the three datasets over each
temporal window.
*Upper layer*
The profile of the 10-year rolling trend of the OHC evaluated based on the three datasets was similar in shape; they
captured most of the peaks and troughs pretty consistently. There was a better agreement among the data for the Indian
Ocean (Fig. 5d) compared to that in the other two basins (Fig. 5b, c), however, notable differences were still observed
in this shallow layer of the Indian Ocean. The rolling trend for the global ocean, estimated from EN4, was mostly
positive, except at the beginning of the 1960s and the end of the 1970s and the 1980s (Fig. 5a). The OFES1 showed a
cooling trend in the global ocean before ~1990; it then indicated a larger warming trend compared to that estimated
from the other two datasets. The OFES2 generally had a better agreement with EN4 for the global ocean, however,
the warming trend was significantly smaller than that estimated using EN4 from the late 1960s to ~1990. Since the
beginning of the 1990s, the disparity in the trend between OFES2 and EN4 was significantly reduced, although the
OFES2 still showed a consistently weaker warming trend. This improved agreement may be attributed to two factors.
First, after running the simulation for approximately 30 years, the OFES2 is expected to have developed better spun-

up and, therefore, the associated results were expected to be closer to the actual state. Second, it is also possible that the accuracy of the EN4 data increased as more observational data were included, given that oceanographic observations have increased significantly since the 1990s (e.g., satellite-based SST measurements and in-situ temperature measurements).

Among the differences observed between the three datasets, the three extreme trend peaks at approximately 1970, 1980, and 2000 (Fig. 5a) were particularly prominent, with remarkable differences between OFES and EN4, indicating some limitations of unconstrained numerical models in the reproduction of strong climate events. The OFES1 was closer to EN4, showing significant warming in the Indian Ocean in the 2000s, whereas OFES2 showed a relatively weaker warming trend. The second better agreement between the three datasets was reached for the Atlantic Ocean.

It was evinced that HV has dominated the 10-year rolling trend in all basins (Fig. 5e–h), and the major differences between the three datasets resulted from the differences in the HV component. In addition, there was an generally out-of-phase relationship between the HV and SP trends in the global ocean and the Pacific Ocean. This correspondence between the HV and SP is expected for typical stratification in subtropical regions (Häkkinen et al. 2016), with warm and salty water overlying cold and fresh water. The OFES1 and OFES2 provided quite similar results for the simulation of spiciness, particularly in the individual basins (Fig. 5i–l).

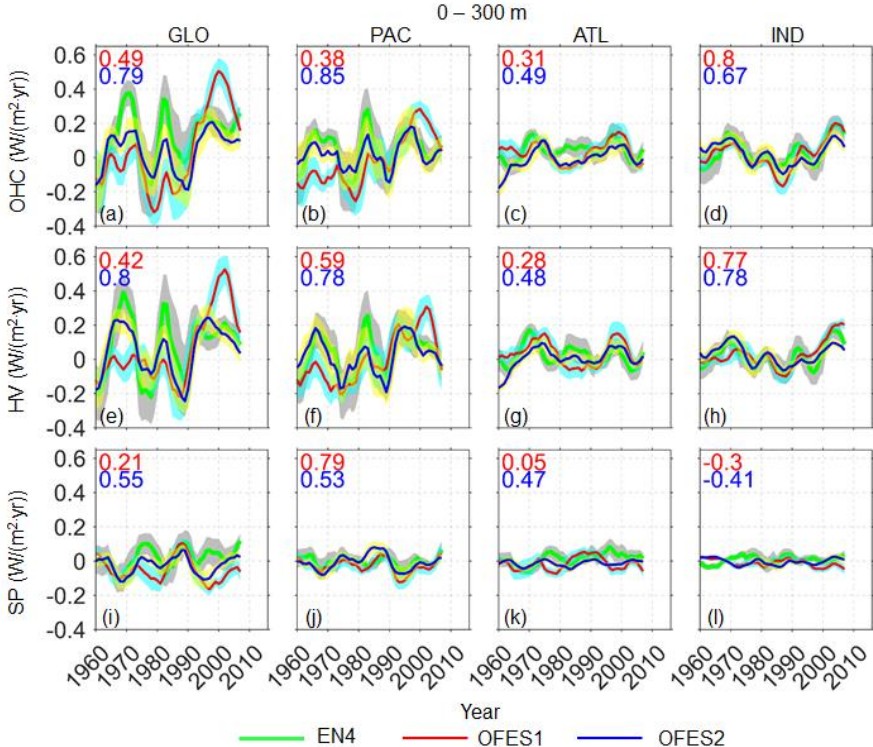

**Figure 5.** Temporal evolution of the 10-year rolling trends in the global and basin OHCs (**top row**), HV (**middle row**) and SP (**bottom row**) in the upper layer (0–300 m), based on the three datasets. Numbers in the top left corners are the correlation coefficients between the EN4 and the OFES1 (red) or OFES2 (blue). The OHC, HV and SP were converted to accumulative heating (W/m$^2$) over the entire surface of the Earth. Thick green line: EN4 (grey shadow: 95% confidence interval); thin red solid line: OFES1 (cyan shadow: 95% confidence interval); thin blue solid line: OFES2 (yellow shadow: 95% confidence interval).

*Middle layer*
The variation in the 10-year rolling trend, evaluated based on OFES1 and EN4 datasets, was found to be similar for
the global (Fig. 6a), Pacific (Fig. 6b), and Atlantic (Fig. 6c) Oceans, however, the latter dataset had a significantly
larger uncertainty. The OFES2 showed a significantly different and generally cooling trend, especially concentrated
in the Atlantic Ocean, consistent with Fig. 3. The origin of the notable cooling trend and its weakening with time
estimated from the OFES2 for the Atlantic Ocean need to be further studied. The cooling trend of the OHC, estimated
from OFES2, was mostly generated from the HV. In the Pacific Ocean (Fig. 6b), the OFES2 consistently showed a
weak cooling trend, however, in the middle and late 1960s and after ~1980, both EN4 and OFES1 showed a warming
trend of similar magnitudes. The results from OFES1 also agreed well with that from the EN4 for the Atlantic Ocean,
i.e., both indicated a weak warming trend for most of the studied period along with a sporadic cooling trend. However,
such agreements could represent the compensation results of the significantly different HV and SP components of
OFES1 and EN4. For example, the EN4 showed a significantly stronger HV warming trend than the OFES1 in the
Pacific Ocean since the early 1990s, however, in the meantime, the EN4 also indicated a stronger SP cooling trend.
In the Indian Ocean, EN4 presented a warming trend over much of the 57 years, whereas the two OFES showed weak
variations and reversals between warming and cooling episodes.

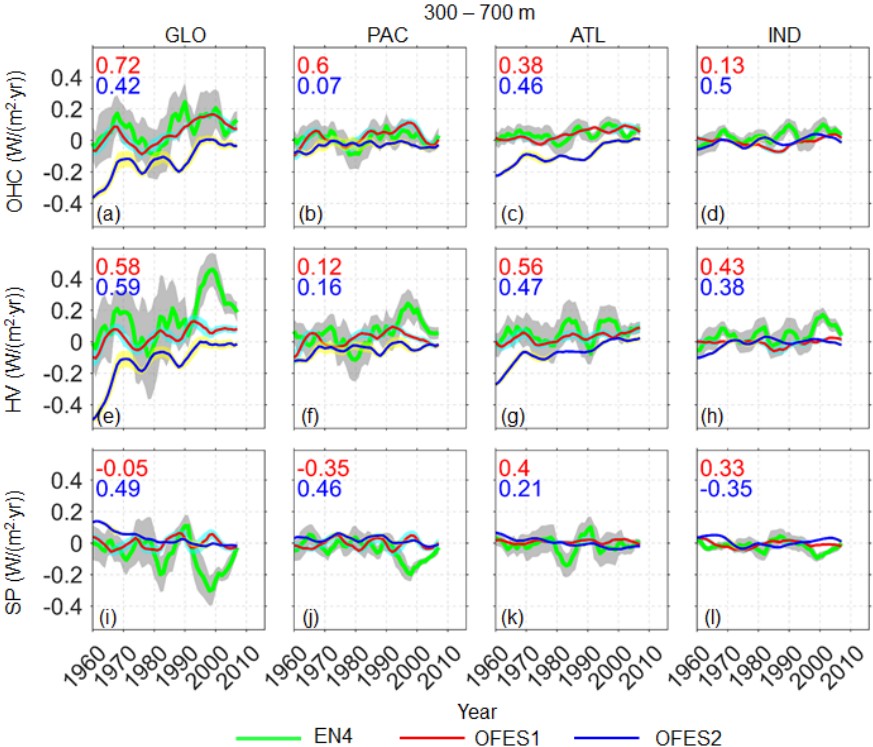


**Figure 6.** As for Fig. 5 but for middle layer (300–700 m).

*Lower layer*
As in the middle layer, the OFES2 differed significantly from the other two datasets by displaying a cooling trend in
the global ocean until approximately 2000 (Fig. 7a). Although OFES2 indicated the appearance of a warming trend in
the global ocean after ~2000, the intensity was significantly lower than that of EN4 and OFES1. The major differences
between the two OFES datasets occurred in the Pacific Ocean (Fig. 7b), and were mostly associated with the HV
component. Despite the good agreement in the OHC trend between the OFES1 and OFES2 for the Atlantic and Indian
Oceans (Fig. 7c, d), their HV and SP components were markedly different, especially in the Indian Ocean (Fig. 7h, l).
The OFES1 and EN4 showed a mostly similar global OHC trend (Fig. 7a); this was because the significant HV and
SP components canceled each other.
To summarize, the OFES2 demonstrated some improvement (better agreement with EN4) over the OFES1 in the
upper layer (above 300 m), but was more of an outlier below 300 m. It is essential to examine the HV and SP
components while investigating the OHC trends because different data products might show mostly similar evolution
of the OHC but substantially different HV and SP.

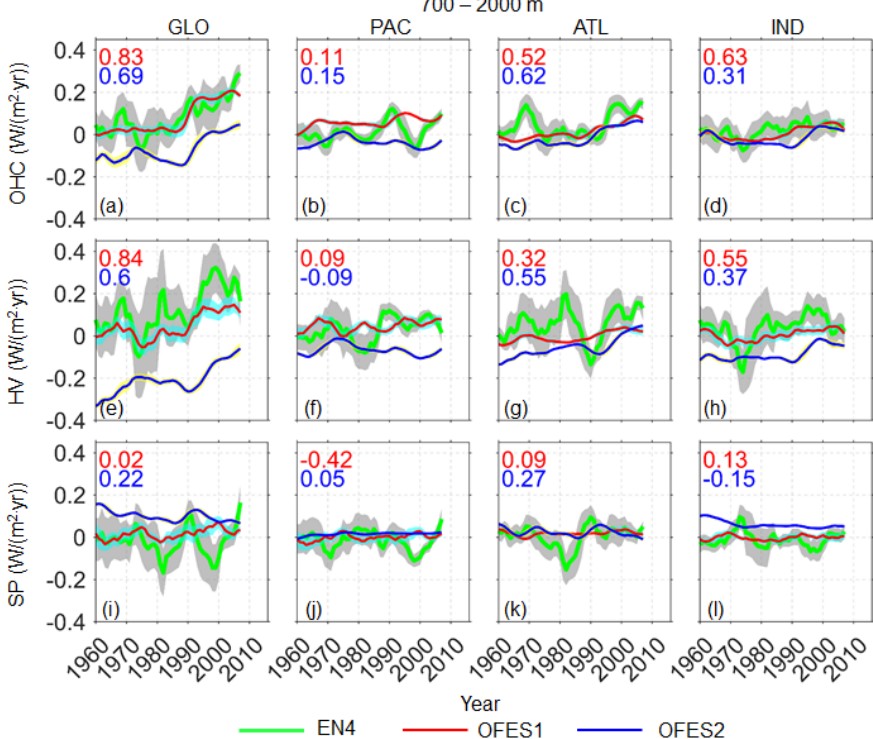

**Figure 7.** As for Fig. 6 but for the lower layer (700–2000 m).

### 3.2 Temporal evolution of the zonal-averaged potential temperature trend

Section 3.1 focused on comparisons of the temporal characteristics of the global and basin-wide OHC, HV, and SP
estimated from the three datasets. Although both similarities and differences were demonstrated, the comparison in
the temporal domain lacked spatial information. In this section, we aimed to understand how these similarities and
differences were distributed in the meridional direction. As a first step, we calculated the 10-year rolling trends in the
zonal-averaged potential temperature for all three datasets (Figs. 8–10). We also calculated the HV and SP components
(Supplementary information, Figs. 1–6).
The complex patterns shown in Figures 8–10 defy easy interpretation; therefore, we have focused on the large-
scale patterns of the observed similarities and differences.

*Upper layer*

In general, a reasonable agreement was observed between the three datasets at latitudes of 30–60° N for both Pacific and Atlantic Oceans (there is no northern high latitude in the Indian Ocean). More specifically, a wave-like cooling patch propagating from approximately 60° N to 30° N was observed from 1960 to the end of the 1970s in the global ocean; this propagation was especially evinced in the EN4 and OFES2 data. In addition, there was a northward propagation of a cooling trend in the 1990s between 30 and 45° N, mainly occurring in the Pacific Ocean. It is reasonable to attribute theses cooling episodes to the volcanic eruptions of Indonesia's Mount Agung in 1963, Mexico's El Chichón in 1982, and the Philippines' Mount Pinatubo in 1991. The two hindcast simulations were able to reproduce these climate events.

Following these cooling events, there were three subsequent warming trends as the ocean surface temperature returned to normal after the aerosols released over several years of volcanic eruptions were completely dispersed. Of these warming trends, the one associated with the El Chichón eruption was the most significant, and there was a clear northward propagation of this significant warming trend from approximately 30° N to the subpolar areas. Interestingly, the contributions of SP to this large-scale warming and cooling episodes were comparable to those of the HV (Supplementary Information, Figs. S1–2), contradicting the general impression that HV is the most dominant contributor to the potential temperature changes. In fact, the abovementioned propagation of the cooling patch from approximately 60 °N to 30° N during 1960–1970 was, to a larger extent, associated with the SP.

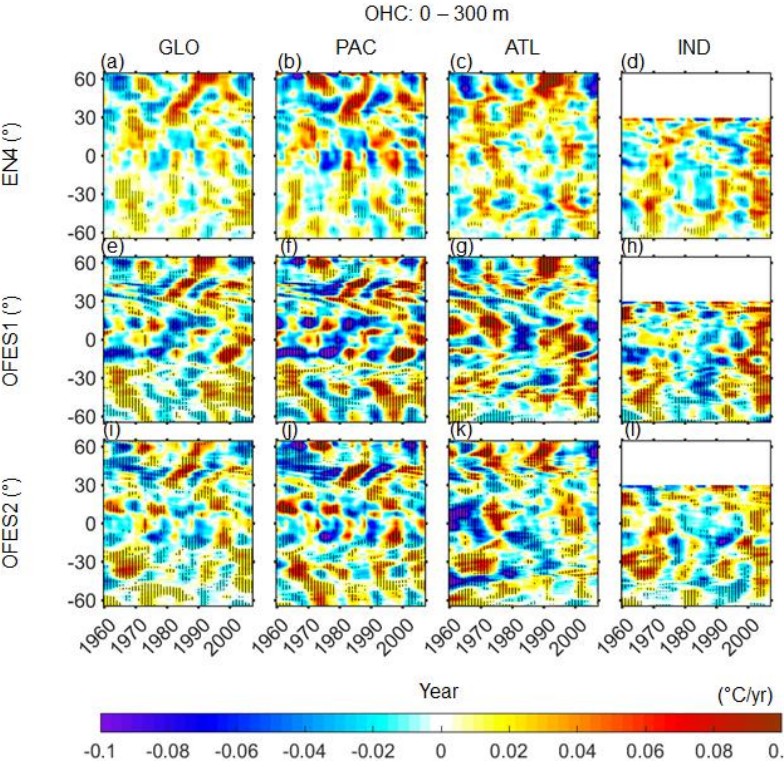

**Figure 8.** Temporal evolution of 10-year rolling trend of the zonal averaged potential temperature change in the upper layer of the ocean (0–300 m). **Left to right:** global, Pacific, Atlantic and Indian Ocean. **Top to bottom:** EN4, OFES1

and OFES2. Horizontal axis: year; vertical axis: latitude. Stippling indicates the 95% confidence level. The HV and
SP counterparts are in the Supplementary Information, Figs. S1–2.

Equatorward of 30°, large differences were observed among the three datasets. Strong cooling was particularly
visible in the OFES1 in the Pacific tropics before around 1990 (Fig. 8f), corresponding to the persistent cooling of the
global ocean and the Pacific Ocean as estimated based on OFEES1 in Fig. 2. The results of OFES2 for the Pacific
Ocean indicated clear differences from the EN4 in the low latitudes before 1980, and then a pattern similar to that of
EN4 was simulated by the OFES2. In the Atlantic tropics, considerable cooling over 1960s was evinced in the OFES2,
which may be the result of poor spun-up in the OFES2. All three datasets captured the Atlantic tropical warming in
the 1970s and from the 1990s to the 2000s, however, the two OFES datasets estimated a stronger intensity than EN4,
especially the OFES1. In addition, OFES1 showed the appearance of significant cooling in the Atlantic tropics during
the 1980s (Fig. 8g). Although a similar contemporary cooling was demonstrated by the OFES2, its cooling center was
shifted several degrees southward. The Atlantic tropical cooling during the 1980s was not notable in  EN4. The OFES2
indicated an approximate 20-year (1960–1980) cooling episode in the vicinity of 45 °S in the Atlantic Ocean (Fig. 8k).
A similar cooling trend existed in the 1960s, but with a relatively weaker intensity in EN4. In the Indian Ocean, the
most significant agreement among the three datasets was observed, particularly the intense warming in the 2000s. In
addition, there were some common cooling patterns observed from the 1980s to the 1990s in all three datasets. It was
shown that the HV accounted for more substantial potential temperature changes than the SP, with the latter generally
counteracting the HV (Supplementary Information, Figs. S1–2).
A general property of the similarities and differences between these three datasets is the fact that a better agreement
was reached in the poleward of 30° than the latitudes equatorward of 30°. A possible explanation for this latitudinal
dependence is that a deeper thermocline at higher latitudes responded less sensitively to the applied wind stress
(Kutsuwada et al., 2019). Kutsuwada et al. (2019) found certain issues with the NCEP reanalysis wind stress that was
used as atmospheric forcing in OFES1 as it generated a significantly shallower thermocline in the tropical North
Pacific Ocean. Therefore, large negative temperature differences were observed when compared to the real
observations along with the data obtained from OFES version forced by the wind stress from satellite measurements
(QSCAT). The authors also claimed that the JRA-55 wind stress had problems similar to that of the NCEP wind.
Indeed, the intense Pacific cooling patches in Fig. 8f were likely generated from the abnormally shallower thermocline
in the tropical Pacific Ocean, consistent with Kutsuwada et al. (2019), although different temporal periods were
considered.

*Middle layer*
In the intermediate layer between 300 and 700 m, the three datasets showed relatively poor agreement compared to
the upper layer. The OFES2 differed from the others by displaying intense cooling before 2000 in the Atlantic Ocean
(Fig. 9k) and a moderate but consistent warming trend in the northern Indian Ocean over almost the entire period (Fig.
9l). In addition, there were large-scale cooling patches in the northern Pacific Ocean (Fig. 9j) and along the Indian
equator (Fig. 9l) from the OFES2, while these cooling patches were not prominent in the other two datasets. These
cooling distributions, obtained from OFES2, further demonstrated the place and timing of the cooling trend shown
Fig. 3, which may be partially attributed to the spin-up issue of the OFES2. Some similarities between the OFES2 and
the other two datasets have emerged in recent decades. For example, similar to EN4 and OFES1, the OFES2
reproduced the marked warming episodes observed in the high latitudes of the northern Atlantic Ocean during the
1980s and 1990s along with the subsequent cooling trend (Fig. 9c, g, k).

461       Upon comparing OFES1 with EN4, both similarities and differences can be discerned. The OFES1 generally agreed

with EN4 in regions located at the north of 30 °N, with some minor differences. However, in the tropics, large
differences were observed between OFES1 and EN4. For instance, the OFES1 indicated that the northern Indian Ocean
was mostly cooling (Fig. 9h), however, EN4 reflected alternate warming and cooling episodes (Fig. 9d). Furthermore,
the intense warming patches of the southern Atlantic Ocean demonstrated by the OFES1 (Fig. 9g) were not apparent
in EN4 (Fig. 9c). These potential temperature changes mainly resulted from the vertical displacement of the neutral
density surfaces, i.e., of the HV component (Supplementary Information, Fig. S3). However, the role of SP cannot be
ignored. This was especially clear in the southern hemisphere of EN4.

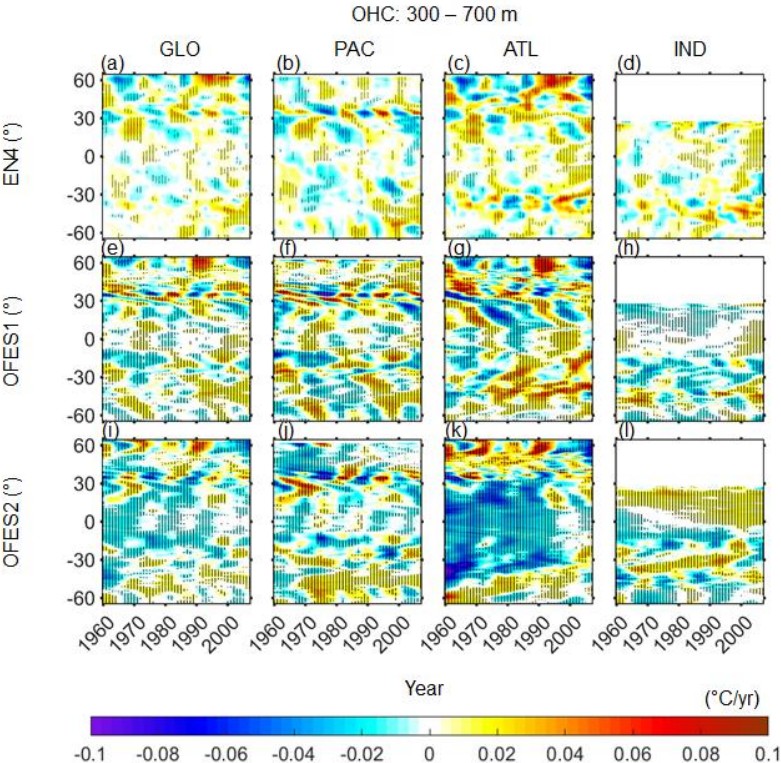


**Figure 9.** As for Fig. 8 but for the middle layer (300–700 m).


*Lower layer*
The northern Atlantic Ocean, especially to the north of 30 °N, dominated the global potential temperature change in
this lower layer (Fig. 10). This was generally more related to SP, especially in the intense cooling patch
(Supplementary Information, Fig. S6). Although the OFES1 data agreed well with EN4 in the northern Atlantic Ocean
(> 30° N), there were considerable differences between OFES1 and EN4. More specifically, OFES1 revealed that
there were intense HV-associated (Supplementary Information, Fig. S5) warming and cooling in the southern Pacific
Ocean during the 1960s and 1970s, however, such trend was not evinced in EN4. In addition, the warming of the
southern Pacific Ocean was much stronger in OFES1 than in EN4 since approximately 1990, which was associated
with the strong SP cooling in the southern Pacific Ocean, as revealed in EN4 (Supplementary Information, Fig. S6).
Moreover, OFES1 demonstrated consistent cooling of the Atlantic tropics, significant warming of the southern
Atlantic Ocean, and intense cooling of the northern Indian Ocean before the middle of the 1990s, which were not
evident in the EN4.
The OFES2 data captured some warming patterns in the southern hemisphere, similar to the OFES1; it also agreed
with the other two datasets in terms of the intense warming patchs in the northern Atlantic Ocean in 1960s and after
~1990. However, the agreement between OFES2 and the others was generally poor. This was most noticeable in the
cooling episode indicated by the OFES2 at the low and middle latitudes for both the Pacific and Atlantic Oceans,
especially the latter. OFES2 showed marked SP variations in the northern Atlantic Ocean (>30 °N), but generally
opposite to that as indicated by EN4. OFES1 indicated moderate SP in a similar warming/cooling pattern to EN4.
To summarize, the two OFES datasets had some good agreements with EN4 for the upper ocean layer, however,
such general agreement was largely confined to the middle-high latitudes. In general, the agreement for the ocean at
lower levels was poor. Specifically, in the middle ocean layer, the OFES1 displayed a generally reasonable agreement
with the EN4 for locations north to 30° N, however, large differences were observed elsewhere. In the OFES2,
intensive cooling patches were simulated, especially in the Atlantic Ocean. Although the spin-up issue may partially
explain the notable differences between the OFES and EN4 data for ocean water below 300 m, other causes might
have also contributed toward the examined differences.

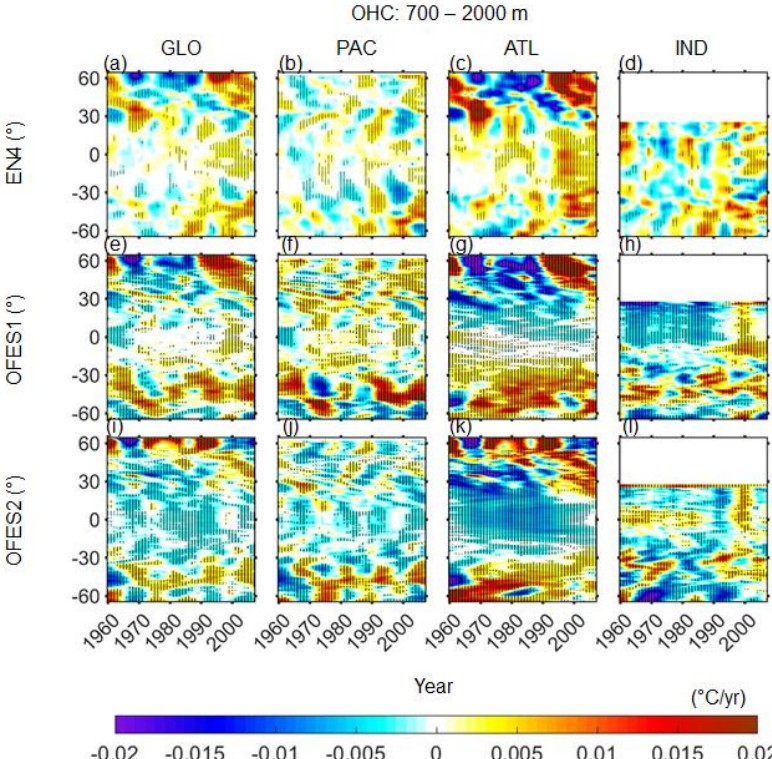


**Figure 10.** As for Fig. 8 but for the lower layer (700–2000 m). Note the different colour scales.

**3.3 Depth-time distribution of potential temperature, HV, and SP trends**

Although we divided the top 2000 m into three layers, some details were lost while considering the averages of individual layers (i.e., the three vertical layers). In this section, we compare the depth-time patterns of the trends with respect to changes in potential temperature ($\Delta\theta_{OHC}$) and its HV ($\Delta\theta_{HV}$) and SP ($\Delta\theta_{SP}$) components (Figs. 11–13).

For the global ocean, the upper ocean layer above 300 m accounted for most of the warming or cooling trends (Fig. 11, left column). EN4 showed warming episodes over most of the investigated period, with only a few cooling episodes as a response to certain distinctive climate events. It can be seen that the volcanic eruptions of Mount Agung and El Chichón had a greater impact compared to the eruption of Pinatubo. The aforementioned strong cooling episode in the upper Pacific layer before 1990, which has been estimated from the OFES1, was initiated at a greater depth in the beginning, and subsequently, it terminated at a shallower depth (Fig. 11e). In the middle and lower layers, moderate warming or cooling trend was observed. Specifically, in EN4, moderate warming has extended to approximately 2000 m, since the early 1990s. The OFES1 showed moderate warming between 500 and 1000 m over almost the entire investigated period (Fig. 11e). Additionaly, it indicated that since the middle of the 1990s, a weak warming trend has extended to 2000 m. The differences in the results of OFES2 relative to the other two datasets are apparent in the global ocean below approximately 200 m, where cooling is the dominant pattern (Fig. 11i); some weak warming patches between 500 and 1000 m are exceptions (Fig. 11i).

In the Pacific Ocean, the OFES2 had a generally reasonable agreement with EN4 above approximately 200 m, whereas the agreement between OFES1 and EN4 was poorer, despite some similar warming or cooling patches. Further below, EN4 showed alternate warming and cooling trends. The OFES1 reflected consistent warming between 500 and 1200 m, whereas the OFES2 estimated a consistent cooling trend below around 200 m, with some exceptions between 500 and 1000 m. Although beyond the scope of this work, the question of why both OFES1 and OFES2 showed relatively consistent warming trends between 500 and 1000 m near the permanent thermocline necessitates further work.

In the Atlantic Ocean, intense warming or cooling extended to deeper regions than in the Pacific Ocean. More specifically, the strong warming trend in the 1980–90s, estimated from EN4, extended to as deep as approximately 750 m. On the other hand, moderate warming trend extended to 2000 m since the middle of 1990s in EN4. The OFES1 well captured the warming trend of the 1970s and 1990s, along with the subsequent cooling period in the 2000s in the upper layer of the Atlantic Ocean, similar to EN4. However, the OFES1 estimated a strong cooling in the 1980s in the upper layer of the Atlantic Ocean, which was not evinced in the EN4. Interestingly, the OFES1 showed downward propagation of a strong Atlantic warming trend from approximately 200 m to 800 m since the early 1980s. Downward propagation of the cooling trend from approximately 600 m to 1800 m before ~1990 was also evinced in the OFES1 data of the Atlantic Ocean (Fig. 11g). Similar to EN4, a moderate warming trend extended to 2000 m since the middle of the 1990s in OFES1. In the case of OFES2, the most prominent pattern that distinguished it from the others was the extensive cooling patchs before ~1990 in the upper and middle layers. In addition, it showed a moderate cooling below 1000 m before 1990. These two extensive cooling patterns in the upper-middle and the lower layers of the Atlantic Ocean, estimated using OFES2, raised the following questions: i) What are the main causes of the two cooling patches exhibited in the OFES2, and ii) Why the cooling patches suddenly terminated at approximately 1990? One possible

reason is the improvement in the reanalysis product of the atmospheric forcing since 1990, especially in the surface
HF and wind stress components, the latter being proved to be essential for subsurface temperature simulations
(Kutsuwada et al. 2019).
In the Indian Ocean, both OFES1 and OFES2 captured the warming trend in the 1960–70s and the 2000s, similar to
EN4. The OFES1 presented an intense cooling in the upper-middle layer during the 1980s; a similar but less extensive
and shallower cooling was also evinced in OFES2. But this cooling patch was significantly less prominent in EN4.
Beneath the upper layer, EN4 presented mostly warming in the Indian Ocean, with a major exception of a cooling
trend in the 1970s. In the two OFES, cooling pattern was more prominent than warming below 500 m, especially in
OFES2. However, between 500–1000 m, warming patches were seen in the 1960s and after ~1990, in both OFES1
and OFES2.

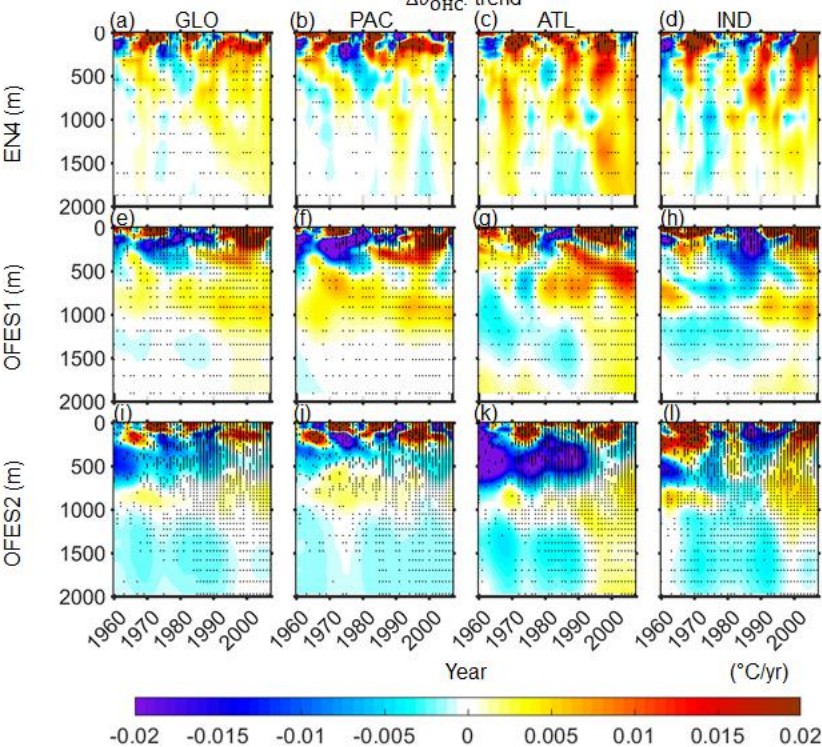

**Figure 11.** Depth-time patterns of the horizontally averaged potential temperature change $\Delta\theta_{OHC}$ for (left to right) the global, Pacific, Atlantic and Indian Oceans. **Top to bottom:** EN4, OFES1 and OFES2. Horizontal axis: year; vertical axis: depth in m.

Upon comparing Fig. 11 with Figs. 12 and 13, it is evinced that to a great extent, the HV components dominated the
OHC variations. For instance, the profound warming and cooling patterns observed in Fig. 11 are mostly associated
with the HV component. The moderate cooling trend observed below 1000 m in OFES2 was also dominantly related
to HV. Although the SP was generally weaker and less important than the HV in accounting for the OHC variations,
its role cannot be ignored. Indeed, intense warming or cooling episodes associated with the SP component were
observed in EN4 in all major basins. The intensified subsurface SP cooling since the 1990s in the Pacific and Indian
Oceans, as indicated by EN4,  has been particularly interesting, which could be associated with a significant increase
in subsurface salinity observations since the 1990s. A possible explanation for the appearance of the intensification of
SP cooling in the Pacific and Indian Oceans, but not in the Atlantic Ocean, is that the Atlantic Ocean has been better
observed than the Pacific and Indian Oceans before the 1990s. Another interesting point with regard to the SP is the
consistent SP warming trend that is observed in OFES2, especially in the Indian Ocean, and not in the other two
datasets.

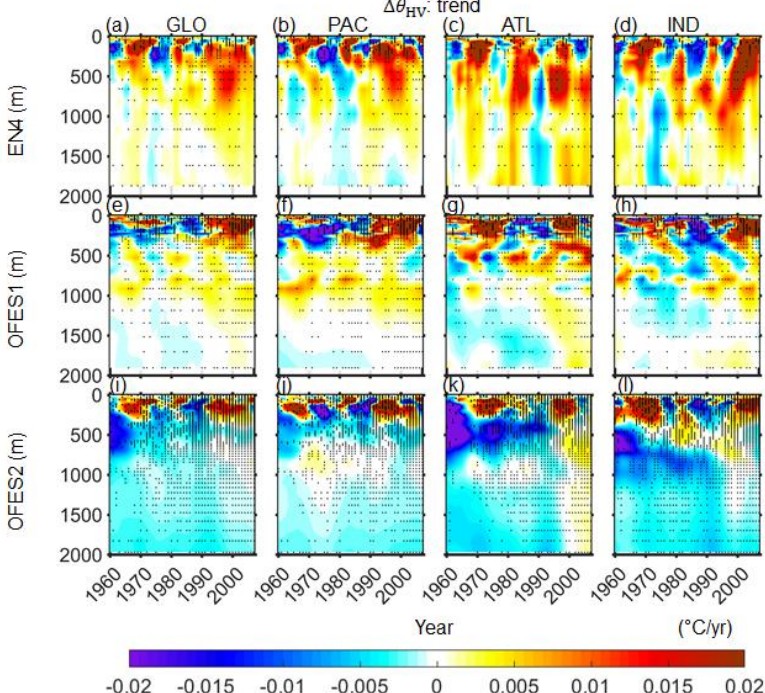


**Figure 12.** Depth-time patterns of the horizontally averaged potential temperature change from the HV component,
$\Delta\theta_{HV}$, for (**left to right**) the global, Pacific, Atlantic and Indian Oceans. **Top to bottom:** EN4, OFES1 and OFES2.
Horizontal axis: year; vertical axis: depth in m.

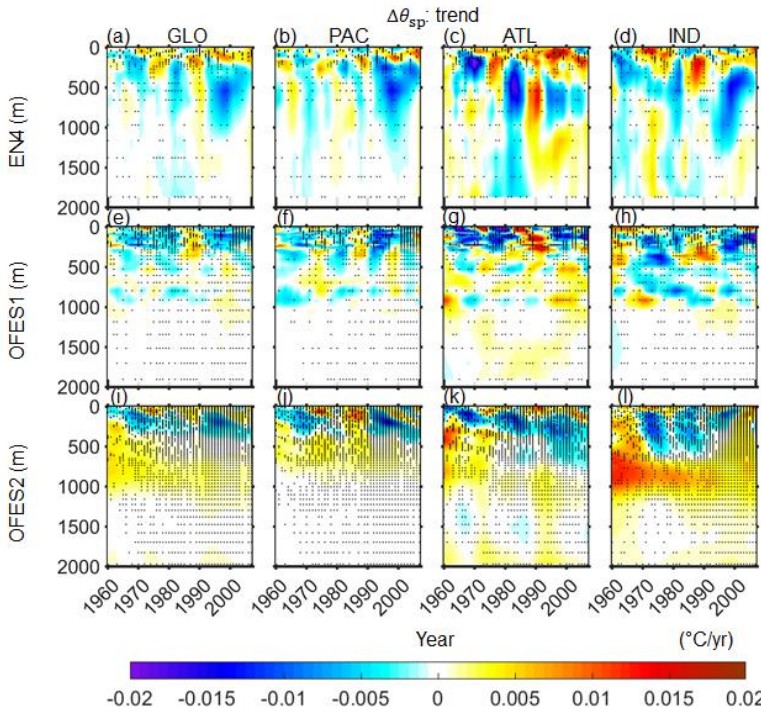


Figure 13. Depth-time pattern of the horizontally averaged potential temperature change from the SP component, $\Delta\theta_{SP}$, for (**left to right**) the global, Pacific, Atlantic and Indian Oceans. **Top to bottom:** EN4, OFES1 and OFES2. Horizontal axis: year; vertical axis: depth in m.

### 3.4 Spatial patterns of the potential temperature, HV, and SP trends

To gain a more detailed understanding of the similarities and differences between the trends of potential temperature estimated from the three datasets, here we have presented the spatial distributions of the potential temperature change ($\Delta\theta_{OHC}$), and its HV ($\Delta\theta_{HV}$) and SP ($\Delta\theta_{SP}$) components in the three ocean layers (Figs. 14–16).

*Upper layer*

Warming was almost ubiquitous in EN4 (Fig. 14a) and was particularly strong in the northern Atlantic Ocean and the Southern Ocean. These two warming hotspots are expected from both theories and models. More specifically, the shallow ocean ventilation in these two regions could generate faster warming than the global average (Banks and Gregory 2006; Durack et al. 2014; Fyfe 2006; Talley 2003). Major cooling appeared in the western Pacific equator, along the North Pacific Current, in the southeastern Pacific Ocean, parts of the Argentine Basin, and the southern Indian tropics. All of these cooling regions accounts for a small fraction of the global ocean. Similar to EN4, both OFES datasets showed significant warming in the subtropics, the high latitudes of the northern Atlantic Ocean, and the Arabian Sea of the Indian Ocean. In addition, the OFES1 was similar to EN4 in terms of cooling along the North Pacific Current. Despite these similarities, large differences exist between the three datasets. The most significant difference was observed in the Pacific tropics. Although EN4 indicated the presence of a zonal band of cooling in the Pacific tropics, this zonal band, when estimated using the OFES1 and OFES2 data, was much stronger in intensity

and more extensively stretched. It was mainly related to the HV component, especially in the case of OFES1. This
strong cooling pattern in the vicinity of the equator was likely generated because of the poor qualities of the
atmospheric wind stress over certain periods. As mentioned earlier, Kutsuwada et al. (2019) demonstrated that the
NCEP wind stress used for forcing the OFES1 data generated a significantly shallower thermocline in the north Pacific
tropical area, and therefore, negative differences were observed relative to the observations. In the northeast of the
Pacific Ocean, the OFES2, but not the OFES1 and EN4, showed a patch of intense cooling, corresponding to the
cooling pattern in the 1960–70s (Fig. 8j). The OFES2 also showed a couple of large cooling areas in the Atlantic
Ocean (Fig. 14g). In the Indian Ocean, the OFES1 and OFES2 datasets indicated the presence of a patch of intense
cooling in the southern Indian tropic and in the Indian sector of the Southern Ocean. Significant cooling also appeard
in the western part of the north Indian Ocean in OFES1.
The decomposition of the changes in potential temperature into HV and SP components showed that the warming
trend, estimated using EN4, was largely the result of isopycnal deepening (HV) in the subtropics. This is consistent
with the finding that the subtropical mode water (STMW) is the primary water-mass accounting for global warming
(Häkkinen et al., 2016), as discussed later. The SP was generally weaker than the HV and tended to counteract the HV
warming, especially in the subtropics. This dampening effect can be easily understood from Fig. 1 of Häkkinen et al.
(2016). For example, in a stratified ocean with warm and salty water overlying cold and fresh water, which is typically
found in the subtropics, complete warming of one water parcel can be considered as the vector sum of warming and
salination component, manifested as a transition from its original isopycnal to a new isopycnal(HV part) and a cooling
and freshening component along the original isopycnal (SP). Two major exceptions of this cancellation between HV
and SP were the northern Atlantic subtropics and the southern Indian Ocean in EN4, where HV warming was mostly
accompanied by the SP warming. The SP warming in the northern Atlantic subtropics was generated owing to a
substantial increase in salinity through evaporation (Curry et al., 2003; Häkkinen et al., 2016). Similarly, we found
widespread positive SP warming in most of the Indian Ocean in EN4, except west to southwest Australia. This SP-
related warming in the northern Indian Ocean dominantly controlled the potential temperature change in EN4,
especially in the Arabian Sea. The most significant SP warming, however, was found in the Indian sector of the
Southern Ocean (may be related to the salination of the Southern Ocean), southern subtropics of the Atlantic Ocean,
and Labrador Sea (Fig. 14c).

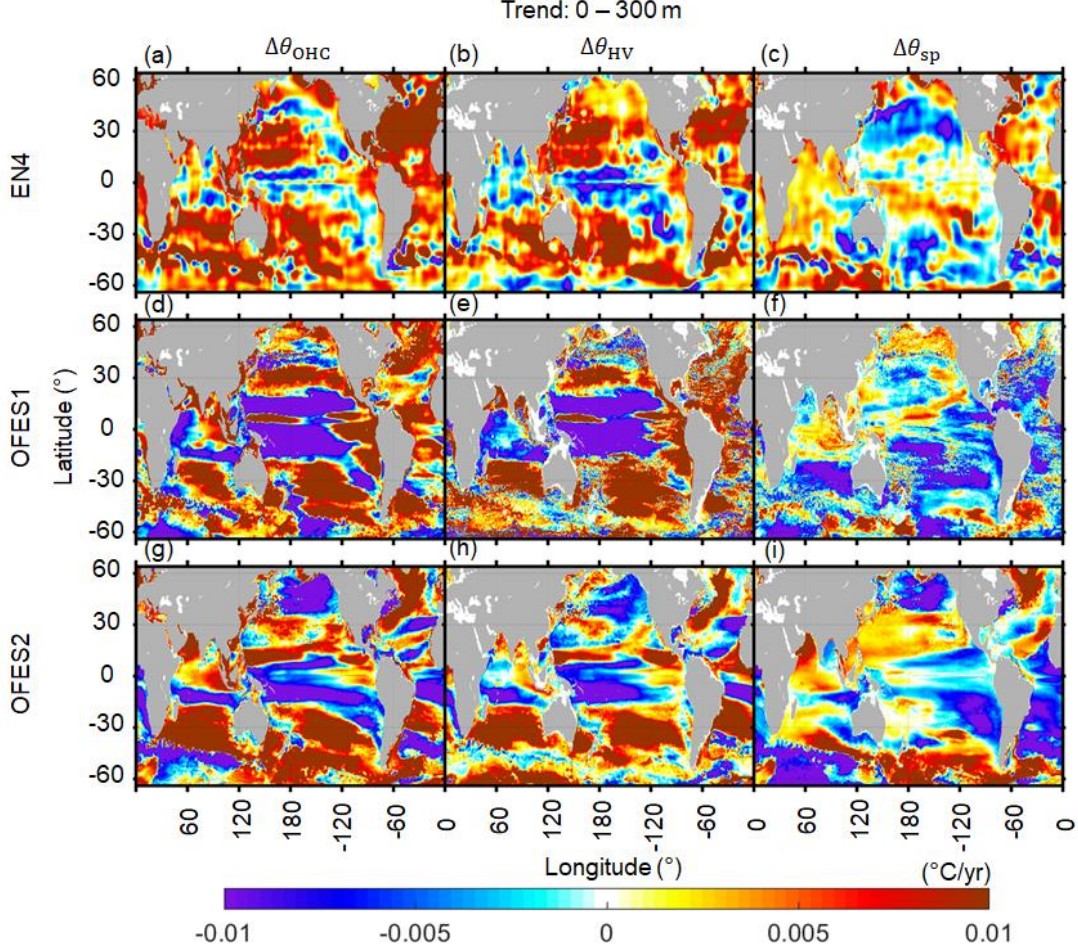

**Figure 14.** Spatial distributions of $\Delta\theta_{OHC}$ (**left column**), $\Delta\theta_{HV}$ (**middle column**) and $\Delta\theta_{SP}$ (**right column**), 1960–2016, in the top ocean layer (0–300 m). **Top** to **bottom**: EN4, OFES1 and OFES2.

Comparing the HV components in the three datasets showed that the two OFES simulations were able to reproduce some subtropical HV warming patterns, although less accurately in the northern hemispehre. The strong and extensive equatorial cooling in the Pacific and Indian Oceans was largely associated with variations in HV in the two OFES datasets.

The SP in the OFES1 was similar to EN4 in the northern subpolar region of the Pacific Ocean, parts of the northern Pacific subtropics, the Labrador Sea, and parts of the northern Indian Ocean. The SP, estimated using OFES2, was similar to the estimates from the EN4 in the Labrador Sea and the western Indian Ocean. In general, however, no common patterns were observed in most of the global oceans. Neither of the OFES datasets captured the SP warming in the western part of the northern Atlantic subtropics. The OFES2, but not EN4 and OFES1, indicated moderate SP warming in the North Pacific subtropics and intense SP warming in the Pacific sector of the Southern Ocean, respectively. The improvements in SP determined based on the OFES2 dataset over that from the OFES1 in the Arabian and Indonesian seas, and not in the Bengal Bay, is partly consistent with **S2020**. The authors demonstrated a smaller bias in the water-mass properties of the Arabian and Indonesian seas, however, a large salty bias remained in the Bengal Bay in the OFES2.

In Fig. 2, we show that the SP, estimated using EN4 and OFES2, was largely similar in the upper layer of the Pacific
Ocean. However, the spatial distributions of the SP component in the Pacific Ocean were seldom similar between EN4
and OFES2. In other words, the time series of a basin-wide quantity hides many details.

*Middle layer*
EN4 showed that the cooling of the ocean was mostly concentrated in the southern Pacific subtropics and the region
associated with the Kuroshio (Fig. 15a). Clear warming trend was observed, accompanied by sporadic cooling patches
in the rest of the global ocean, especially over most of the Atlantic Ocean, in the northern Indian Ocean, and along the
Antarctic Circumpolar Current (ACC) path of the Southern Ocean. The OFES1 dataset could reproduce some warming
patterns in the northern Pacific Ocean, the bulk of the Atlantic Ocean, the eastern part of the northern Indian Ocean,
and parts of the ACC path. However, notable differences were found between OFES1 and EN4. Among these
differences, the most prominent is the intense cooling in the southern Indian Ocean as estimated from OFES1. In
addition, strong cooling patches were also found in the southern Pacific tropics, west to central-south America, in the
northern Atlantic subtropics, in the Arabian Sea, and along parts of the southern edge of the ACC in OFES1. The
pattern in the OFES1 Pacific Ocean clearly appears as zonal bands. Consistent with Fig. 3, intense cooling was
simulated by OFES2 for all major basins, with the most prominent being in the Atlantic Ocean. Large-scale warming
patterns were found in the Kuroshio region, in the southern Pacific and Indian subtropics, in the northern Atlantic
Ocean (north of 35° N), in the western part of the northern Indian Ocean, and in the Pacific and Atlantic sectors of the
Southern Ocean. In general, there were apparent differences between the three datasets when the bulk of the global
ocean was considered. The above 700 m is relatively well observed, especially in the Atlantic Ocean (even back to
1950–60s, Häkkinen et al., 2016). Therefore, it is likely that the OFES2 dataset was an outlier at the analyzed multi-
decadal scale, and there could be some potential problems in the OFES1, for example, in the southern Indian Ocean.
Interestingly, EN4 suggested that HV warming was almost ubiquitous in the middle layer (Fig. 15b), especially in
the southern hemisphere, which is consistent with the warming shift toward the southern hemisphere (Häkkinen et al.,
2016). Correspondingly, SP cooling also occupies most of the global ocean (Fig. 15c), with a similar southern shift,
the most prominent being around the east and western regions of Australia. Major SP warming patches were found in
the Sea of Okhotsk, north of the Gulf Stream, in the Arabian Sea, and along the southern edge of the ACC. These
regions are generally associated with strong variations in salinity. Comparing HV and SP estimated based on EN4 and
OFES1 dataset showed that the OFES1 captured some warming patterns in the Pacific and Atlantic, but not in the
Indian subtropics. The agreement of HV for the southern Pacific, Indian tropics, and the Southern Ocean was mostly
poor. In the case of SP, the OFES1 reproduced intense SP cooling in western Australia and the southern Pacific
subtropics, similar to EN4, despite its smaller coverage. However, OFES1 showed almost opposite trends of SP over
most of the global ocean. In OFES2, both HV and SP were strong, however, the basin-wide cooling was mainly
generated as a result of HV. Overall, the OFES2 dataset had a reasonable agreement with EN4 in the southern
subtropics (Pacific and Indian Oceans) in terms of HV. It also had a common HV warming patch in the northern
Atlantic Ocean (north to 35° N) as EN4. With regard to SP, the OFES2 was similar to EN4 in displaying SP warming
in the Arabian Sea and parts of the southern edge of the ACC. In addition, it also captured SP cooling in the eastern
Pacific Ocean, along the Gulf Stream path, west of Australia. Except for these similarities, however, OFES2 dataset
was generally not consistent with that of EN4 in terms of SP.

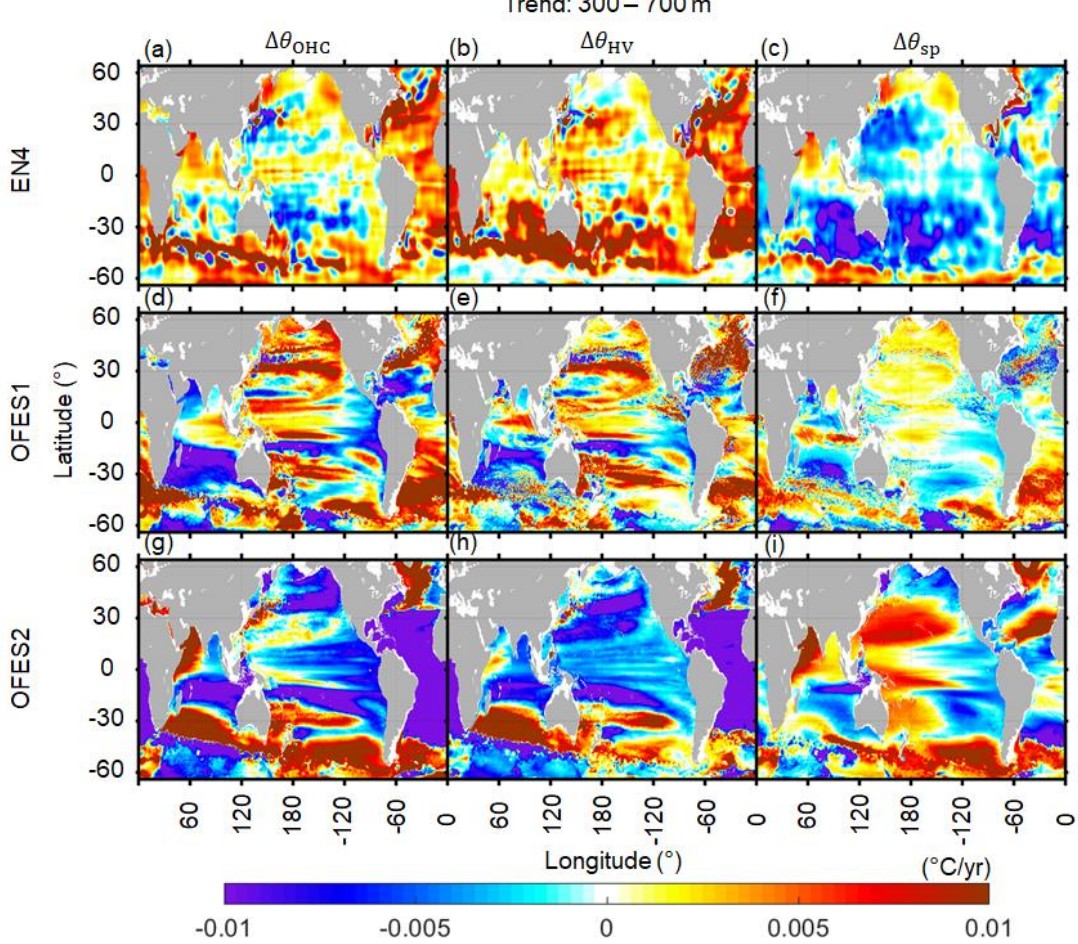


**Figure 15.** As for Fig. 14 but for the middle layer (300–700 m).

*Lower layer*
In general, the warming and cooling intensities were significantly weaker in the lower layer compared to that in the
top two layers, which is consistent with several previous findings that more heat was stored in the upper 700 m than
at greater depths (Häkkinen et al., 2016; Levitus et al., 2012; Wang et al., 2018; Zanna et al., 2019). EN4 showed
widespread warming patches in the Southern and Atlantic Oceans, and three large zonal bands of cooling in the
southern subtropics of the Pacific and Indian Oceans, and in the northern subpolar region of the Atlantic Ocean (Fig.
16a). Similar to EN4, the OFES1 dataset reflected warming along the northern edge of the ACC and in the southern
Atlantic Ocean, but the intensity of warming was much stronger for OFES1 than in EN4 (Figs. 16a, d). OFES1
reflected moderate warming over almost the entire Pacific Ocean, which was not the case in EN4. Significant
differences between OFES1 and EN4 were also found in the northern Atlantic Ocean, where the OFES1 showed
extensive cooling compared to the moderate warming in EN4. OFES1 demonstrated strong cooling in the Arabian
Sea, which is in contrast to negeliable varaitions the Arabian Sea obtained from the EN4. To some extent, the OFES2
was similar to the other two datasets in showing warming along the northern edge of the ACC and in the southern
Atlantic Ocean, south to 30 °S (Fig. 15g), despite the differences in the intensity of warming. It also showed cooling
in the low and middle latitudes of the Atlantic Ocean, similar to OFES1 but opposite to EN4. The bulk of the Pacific
Ocean was shown to be cooling in the OFES2 (Fig. 15g), which was almost opposite to the OFES1 results (Fig. 15d)
and similar to EN4 only in parts of the southern Pacific subtropics (Fig. 15a). Moreover, OFES2 reflected intense and
widespread cooling in the Indian sector of the Southern Ocean.

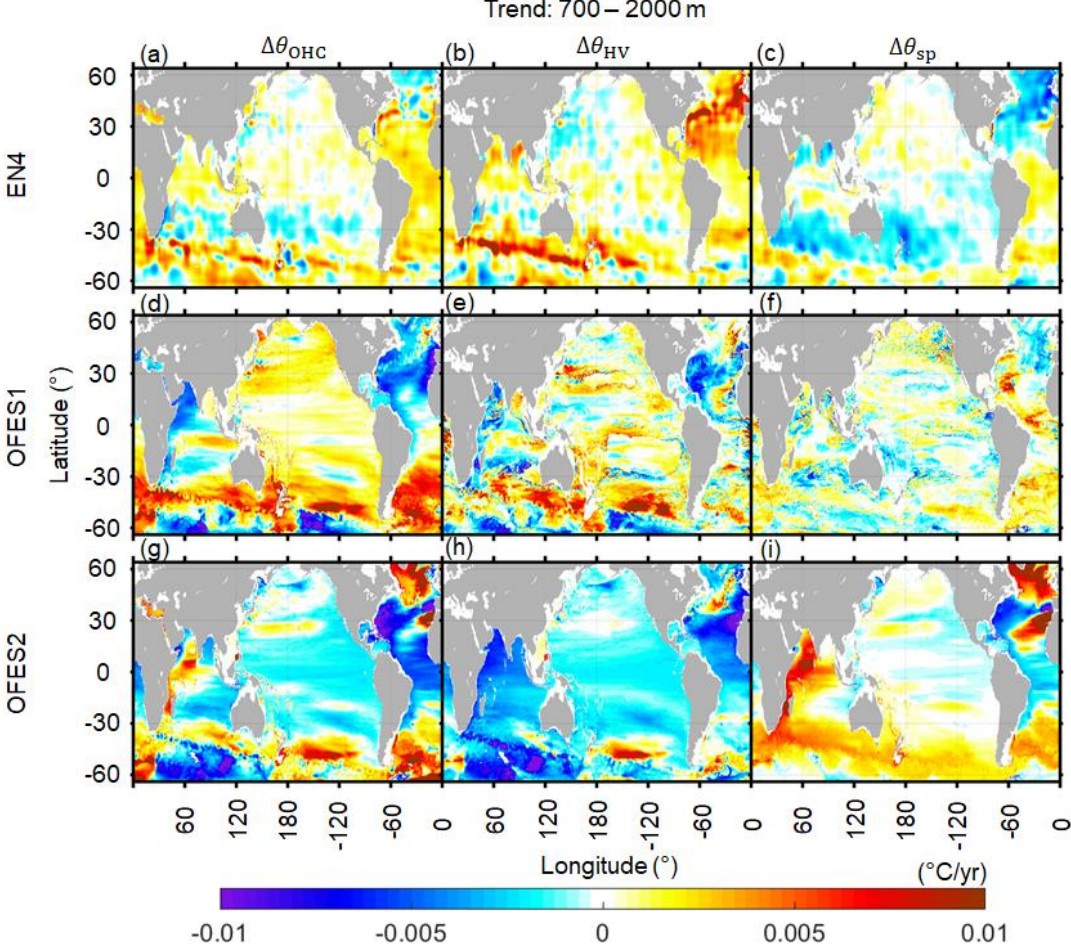


**Figure 16.** As for Fig. 14 but for the lower layer (700–2000 m).

In NE4, there was intense HV warming along the northern edge of the ACC in the Indian and Pacific Oceans, and in
the northern Atlantic Ocean (Fig. 16b), which largely accounted for the total potential temperature variations. HV
warming was generally accompanied by SP cooling (Fig. 16c). Moderate HV and SP warming coexist in the northern
Atlantic tropics and the southern Atlantic Ocean in EN4. We found that the OFES1 captured the HV warming pattern
along the northern edges of the ACC, which to some extent, is consistent with the results from EN4. However, there
were remarkable differences in OFES1 results from those of EN4, particularly in the northern Atlantic and Indian
Oceans. In terms of SP, there were some similarities between the OFES1 and EN4; for example, they both had SP
cooling and warming in the northern and southern Atlantic Ocean, respectively. Among the three datasets, OFES2
showed the most extensive and strong HV-associated cooling, except for a patch of HV warming in the Pacific sector
of the Southern Ocean, which was also observed in the other two datasets. The OFES2 estimated intense SP warming
in the Southern Ocean, the western Indian Ocean, and the northern Atlantic subpolar regions. A large-scale patch of
abnormally strong SP warming, associated with the Mediterranean Overflow Water (MOW), was also observed. This
extremely strong SP warming, associated with MOW, is likely the result of the unrealistic spreading of the salty
Mediterranean overflow reported in **S2020**.
Besides the above-discussed multi-decadal linear trends, we have demonstrated that (not shown here) the significant
differences between the two OFES datasets and the EN4 were significantly reduced if the period between 2005 and
2016 is considered, during which the two OFES were argued to be well spun-up (**S2020**). In addition, over this 12-
years period, the spatial pattern of the OFES2 showed some improvements over the OFES1 for the upper and middle
layers, however, it was not necessarily true for the lower layer when EN4 was used as a reference. Is this better
agreement a result of better spun-up or it was generated owing to improvements in the reanalysis product of the
atmospheric forcing for the two OFES data? This interesting question requires further exploration in the future.

### 3.5 Trends of HV and SP in the neutral density domain

Plotting the HV and SP components in neutral density coordinates provides useful information to analyze the warming
and cooling from the perspective of water-mass. Following Häkkinen et al. (2016), we calculated the linear trend of
the zonal-averaged sinking of the neutral density surfaces in each major basin over 1960–2016 (Fig. 17). We also
calculated the zonal-averaged  SP-related warming or cooling along the neutral density surfaces (Fig. 18).
Our results, based on the EN4 dataset, were similar to those of Häkkinen et al. (2016), who used an earlier version
of EN4 dataset (i.e., EN4.0.2) and considered the period from 1957 to 2011. More specifically, our EN4 results showed
that bulk HV warming (deepening of neutral density surfaces) was associated with water-mass of over 26 $kg/m^3$, and
was mainly concentrated south of 30° S, from the ventilation region at high latitudes to the subtropics. There was one
exception in the Atlantic Ocean, where deepening of isopycal heavier than 26 $kg/m^3$ occurred at all the considered
latitudes. The concentrated warming in the northern Atlantic Ocean was attributed to phase change of the North
Atlantic Oscillation (NAO) from negative in the 1950–60s to positive in the 1990s (Häkkinen et al. 2016; Williams et
al. 2014). As explained by Häkkinen et al. (2016), the significant deepening of neutral density surfaces was associated
with the subtropical mode water (STMW, $26.0 < \sigma_0$ ($kg/m^3$) $< 27.0$) and the Subantarctic Mode Water (SAMW, $26.0$
$< \sigma_0$ ($kg/m^3$) $< 27.1$). These vertical displacements of neutral density surfaces probably resulted from heat uptake via
subduction, which subsequently might have spread from these high-latitude ventilation regions. The large vertical
deepening of the STMW and SAMW had subsequently pushed the Subpolar Mode Water (SPMW, $27.0 < \sigma_0$ ($kg/m^3$)
$< 27.6$) and the Antarctic Intermediate Water (AAIW, $27.1 < \sigma_0$ ($kg/m^3$) $< 27.6$) further down. However, as the vertical
displacement of the STMW/SAMW was larger, its volume would have increased, and the volume of the underlying
SPMW/AAIW decreased (Häkkinen et al., 2016). Besides these significant sinking of neutral density surfaces, there
was generally a shoaling pattern of lower density ($\sigma_0$ ($kg/m^3$)) ranging from 24 to 26, which was mostly concentrated
between the equator and 30° S. To a large extent, this shoaling occurred in the central water, for example, in the South
Pacific Central Water (SPCW).
In this study, we have not focused on the detailed mechanisms of warming from the perspective of water-mass, as
has been done in previous studies (Häkkinen et al., 2016). Instead, we have focused on the differences between the
three datasets with respect to the trends of HV and SP.
It can be seen that along the surfaces of the Pacific and Indian Oceans, there was a general appearance of HV warming
in almost all three datasets. In the Atlantic Ocean, however, the EN4 estimated a sea surface cooling south of 30° S
and in the northern tropic; the OFES2 also estimated a cooling trend near the surface of the Atlantic tropics. In contrast
to both EN4 and OFES2, OFES1 showed an intense HV cooling pattern along the Atlantic surface between 30 and
50° N (Fig. 17e).

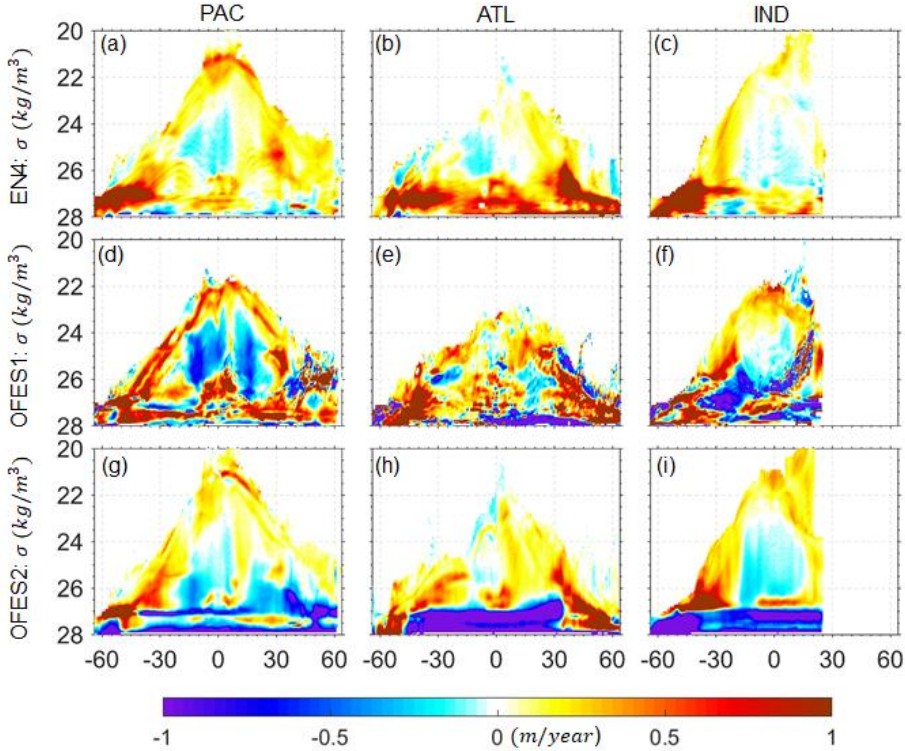


**Figure 17.** Linear trends in the zonal-averaged sinking of the neutral density surfaces in the Pacific (**left column**),
Atlantic (**middle column**) and Indian (**right column**) Oceans. **Top to bottom:** EN4, OFES1, OFES2. Positive values
mean deepening of the neutral density surfaces. The calculation was for the water above 2000 m.


South of 30° S, EN4 detected large downward movements, associated with the STMW, SAMW, and AAIW in all
three basins. In the case of OFES1, the dominant pattern in the three basins was sinking, however, it was surrounded
by shoaling patches; larger differences from the EN4 were found in the OFES2, which showed significant and
extensive shoaling patterns, especially in the Atlantic and Indian Oceans. The almost opposite trend in the vertical
displacements of the neutral density surfaces between the OFES2 and the observation-based EN4 may indicate that
the changes of properties of water-mass simulated in the OFES2 were unrealistic, at least at this multi-decadal scale.
In the ocean interior between 30°S and 30° N, the OFES1 presented shoaling patterns in the Pacific and Indian
Oceans, however, such shoaling pattern was not prominent in the Atlantic Ocean. Although the shaoling patterns in
the Pacific and Indian Oceans were also evinced in EN4, their magnitude was generally weaker. The OFES2 had better
agreement with EN4 for the shoaling pattern in the southern Pacific subtropics. OFES2 also captured shoaling in the
Indian Ocean, with similar coverage, however, the intensity was generally stronger. Shoaling in the southern Atlantic
subtropics was not prominent in OFES2, similar to the OFES1, but different from the EN4.
In the north of 30° N, EN4 detected widespread sinking, particularly in the northern Atlantic Ocean. This strong
sinking in the northern Atlantic Ocean originated mainly from SPMW and STMW. In the EN4 Pacific Ocean, there
were certain shoaling patches, which were related to the North Pacific Intermediate Water (NPIW). In OFES1, the
pattern was filled with both sinking and shoaling patches, which defies easy interpretation. However, an apparent
outlier of OFES1was the intense shoaling in the northern Atlantic Ocean (mostly below 700 m (Figs. 14–16)), which
is the opposite of EN4. The shoaling of neutral density surfaces in the OFES2 Pacific Ocean, north to 30° N, was even
more prominent than that in the OFES1. The OFES2 had a better agreement with EN4 in terms of the sinking patterns
in the Atlantic Ocean north of 30° N.

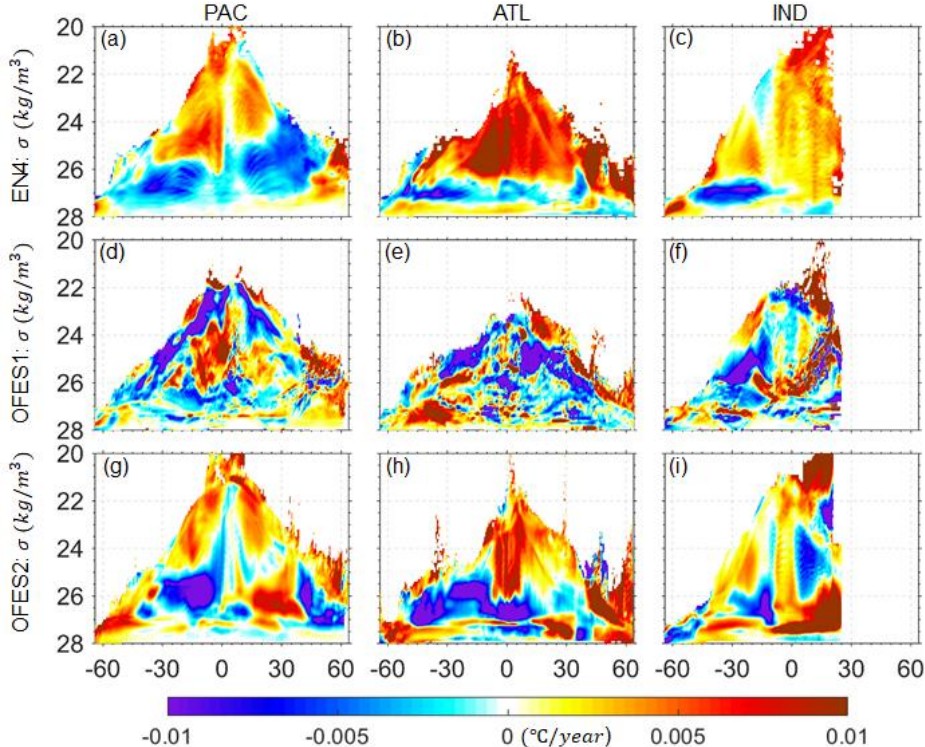


**Figure 18.** Linear trends in the zonal-averaged warming or cooling along the neutral density surfaces in the Pacific
(**left column**), Atlantic (**middle column**) and Indian (**right column**) Oceans. **Top to bottom:** EN4, OFES1, OFES2.

The major SP warming episodes determined by EN4 in the Pacific Ocean were associated with STUW and Pacific
Central Water (PCW) in the low and middle latitudes, with a shift toward the southern hemisphere. The northern high-
latitude SP warming was mainly related to the Pacific Subarctic Intermediate Water (PSIW). The two SP coolings
were generated from the STMW, accompaning the isopycnal deepenings in Fig. 17(a). HV warming/ SP cooling was
particularly typical in the subtropical regions, and HV/ SP warming was typical in the subpolar regions, more details
of which are presented in Häkkinen et al. (2016). An extremely strong SP warming trend occurred in the Atlantic
Ocean, resulting from salination via evaporation. In the southern Atlantic Ocean, the pattern of SP cooling was mostly
accompanied by the sinking of the STMW.
The SP pattern determined from the OFES1 dataset was quite noisy, and generally had a poor agreement between
OFES1 and EN4 in terms of SP warming, which is likely to result from some issues of simulation of salinity in the
OFES1. As shown in **S2020**, OFES1 was not capable of simulating salty outflows, for example, the outflow through
the Persian Gulf into the Indian Ocean. There were notable improvements in the salinity fields of OFES2 over OFES1,
which has been mainly attributed to the inclusion of river runoff and sea-ice, however, some issues associated with
poor performance in the simulation of the MOW remained. Overall, the SP warming pattern in the density coordinate
was significantly improved in OFES2 compared to OFES1. However, upon combining Figs. 14–16, it is evinced that
the similarities in SP estimation between the OFES2 and EN4 dataset were confined to a small fraction of the global
ocean, mainly in the upper and middle layers of the Labrador Sea, northern Indian Ocean, and Southern Ocean. In
addition, the simulations by OFES2 shared similarities with those of EN4 in showing a patch of SP cooling in the
western part of the northern Atlantic subtropics.

**3.6 A basin-wide heat budget analysis**
The primary processes controlling the oceanic thermal state include the net surface HF, ZHA and MHA in the
horizontal direction, and VHA and VHD in the vertical direction (Fig. 1b). Lateral heat diffusion was not considered
here because it was found to play a minor role in our analysis (not shown). Because our focus is on the global and
basin-wide OHC in the three vertical layers, we calculated and compared the inter-basin heat exchange, and the VHA,
integrated over each basin from 1960 to 2016. No vertical heat diffusivity data were available from OFES1. In addition,
the vertical heat diffusivity data from OFES2 were temporarily unavailable because of a security incident when this
manuscript was prepared. This prevented us from directly calculating and comparing the VHD between OFES1 and
OFES2. As an alternative solution, we calculated the residual of the OHC change, along with all associated heat
transport components that contribute to each basin, and used the results as a proxy for VHD. This indirect method
might suffer from some errors; for instance, it includes the impacts of river runoff in the OFES2, however, it can still
provide us with some important information. The calculations are listed in Table 2–4. The related time series of these
surface heat fluxes and heat advection are shown in Supplementary Figs. S7–9.

*Upper layer*
In the Pacific Ocean, the rate of change of the OHC was rather low for both OFES1 and OFES2. The average surface
HF, estimated using the OFES1 dataset, was twice that of OFES2, indicating that heavier heating applies to the OFES1
Pacific Ocean, signifying their differences in atmospheric forcing. Vertically, both datasets indicated a net downward
advection of heat in the Pacific Ocean at 300 m, however, the intensity was much stronger in the OFES1 (different by
approximately 0.7 W/m$^2$), which may be related to their different wind-forcing sources, as the downward heat
advection in the upper ocean was mainly from the wind-driven Ekman pumping in the subtropical gyres. Indeed,
Kutsuwada et al. (2019) claimed that the NCEP wind stress curl was too strong, and had generated the overly strong
Ekman pumping. There was an increase in the eastward heat advection through the water passage between the
Australian mainland and 64° S by 0.150 W/m$^2$ (P3 in Fig. 1a) in the OFES2, in a comparison to the OFES1. Although
the two OFES datasets indicated that the MHA from the Southern Ocean to the Pacific Ocean (P4) had opposite signs,
the relatively small absolute value indicated that this difference was not essential. The Drake Passage (P5) is the major
water passage, through which heat is exchanged between the Pacific and Atlantic Oceans. There was 0.108 W/m$^2$
more heat loss through P5 into the Atlantic Ocean in the OFES1, inferring a stronger ACC from the OFES1 in the
upper ocean. P7 and P8 connect the Pacific and Indian Oceans, and the Indonesian Throughflow (ITF) flows through
P7. The MHA passing through P7 was almost one time stronger in OFES2 than in OFES1, with a difference of 0.637
W/m$^2$. This indicated an enhancement of the ITF simulated by the OFES2. This to some extent, agreed with the results
of Sasaki et al. (2018), who showed that the inclusion of a tidal-mixing scheme resulted in an intensification of the
ITF, noting that the tidal-mixing scheme was implemented in OFES2 but not in OFES1. In addition, the OFES1
demonstrated that more heat was transported westward into the Indian Ocean between Papua New Guinea and
Australia (P8), however, the small absolute heat advection indicated that it was not the major cause of the OHC
discrepancy between OFES1 and OFES2. The net heat advection through the Bering Strait (P9) was rather weak in
both datasets. The indirect calculation of the VHD showed that there was net downward heat diffusion at the depth of
300 m in the Pacific Ocean in both the OFES datasets, although the intensity was much stronger (0.747 W/m$^2$) in the
OFES1.
In the Atlantic Ocean, the OHC increased at an average rate of 0.032 W/m$^2$ in OFES1, however, it decreased by
0.014 W/m$^2$ in OFES2. There was net surface heating in the OFES1 Atlantic Ocean, but minor cooling was evinced
in the OFES2. The two OFES datasets were also profoundly different in terms of VHA at 300 m. Specifically, OFES1
showed a net downward heat advection, and OFES2 showed upward and significantly weaker heat advection. Again,
this difference in the VHA was likely the result of different wind stress datasets in the two OFES, as discussed above.
In a comparison to the OFES2, the OFES1 showed an increase in the heat transported from the Atlantic Ocean to the
Indian Ocean through P1 between South Africa and 64° S by 0.158 W/m$^2$. As mentioned above, more heat was
advected into the Atlantic Ocean through the Drake Passage (P5) in OFES1. Additionally, there was more heat
advected southward from the Atlantic Ocean to the Southern Ocean in OFES1 (P6). The wide passage connecting the
North Atlantic Ocean to the Arctic Ocean (P10) served as the major channel, through which the Atlantic Ocean
exchanged heat with the Arctic Ocean; the two OFES datasets exhibited similar heat loss. All these differences led us
to conclude that the resultingVHD at 300 m differed by 0.411 W/m$^2$ (with stronger upward heat diffusion estimated
by the OFES1).
For the Indian Ocean, the average rate of increase in OHC, calculated by OFES2, was higher than in OFES1 by 0.009
W/m$^2$. The time-averaged surface HF in OFES2 was 0.729 W/m$^2$ lesser than that in OFES1. Both datasets showed a
net downward heat advection, however, the results obtained from OFES2 were approximately two times stronger. The
small difference in the southward heat advection across 64° S ( P2) only affected the OHC in the upper Indian Ocean
to a small extent. In contrast, the differences in the HF, VHA, and MHA associated with the ITF contributed to the
difference and led us to calculate a remarkable difference of 1.898 W/m$^2$ in the VHD at a depth of 300 m in the Indian
Ocean. The enhanced ITF is one of the main contributors to the larger increase of the OHC in the upper layer of the
OFES2 Indian Ocean (Fig. 2).
To summarize, OFES1 estimated a higher surface HF into the major basins. The VHA was generally downward,
indicating the essential role of subtropical Ekman pumping in the heat uptake of the upper ocean layer. The differences
between these two (HF and VHA) contributors were mainly due to the different atmospheric forcing used in the two
OFES datasets, emphasizing the importance of reliable atmospheric forcing products in numerical ocean modeling.
Although the different wind stresses could also produce different lateral advection through P1–P10, the horizontal
heat advection through these passages are largely similar in the two OFES. The most prominent difference in the
lateral heat advection was associated with the ITF, which was to some extent a result of the adoption of a tidal-mixing
scheme. This ITF-related difference and the indirectly inferred VHD suggested the significance of the vertical mixing
scheme in producing the examined differences in OHC.
**Table 2.** Time-averaged OHC increasing rate, surface heat flux (HF) and advection of heat through the major water
passages for the upper layer (0–300 m) of each basin. VHA in this table is at a depth of 300 m. Residual: difference
between the OHC increase and all the heat flux into a basin, approximately the VHD. All quantities converted to $W/m^2$
applied over the entire surface of the Earth. Values smaller than 0.001 are set to 0. Positive means heat gain and
negative means heat loss.

| | | | | | PACIFIC OCEAN (0–300 m) | | | | | |
|---|---|---|---|---|---|---|---|---|---|---|
| | OHC | HF | VHA | P3 | P4 | P5 | P7 | P8 | P9 | Residual |
| **OFES1** | −0.025 | 2.135 | −0.814 | 1.233 | 0.011 | −0.891 | −0.728 | −0.162 | −0.003 | −0.808 |
| **OFES2** | 0.007 | 1.066 | −0.113 | 1.383 | −0.020 | −0.783 | −1.365 | −0.100 | 0 | −0.061 |

| | | | | ATLANTIC OCEAN (0–300 m) | | | | |
|---|---|---|---|---|---|---|---|---|
| | OHC | HF | VHA | P1 | P5 | P6 | P10 | Residual |
| **OFES1** | 0.032 | 0.184 | −0.445 | −0.823 | 0.891 | −0.085 | −0.440 | 0.749 |
| **OFES2** | −0.014 | −0.036 | 0.005 | −0.665 | 0.783 | −0.051 | −0.388 | 0.338 |

| | | | | INDIAN OCEAN (0–300 m) | | | | |
|---|---|---|---|---|---|---|---|---|
| | OHC | HF | VHA | P1 | P2 | P3 | P7 | P8 | Residual |
| **OFES1** | 0.026 | 0.195 | −0.639 | 0.823 | −0.038 | −1.233 | 0.728 | 0.162 | 0.028 |
| **OFES2** | 0.035 | −0.534 | −2.091 | 0.665 | −0.012 | −1.383 | 1.365 | 0.100 | 1.926 |


*Middle layer*
The horizontal and vertical heat transport in the middle layer (300–700 m) of the Pacific Ocean (Tab. 3), estimated by
OFES1 and OFES2, displayed no significant difference. It can be seen that the ITF was weak for this deeper layer,
and the differences in the results from OFES1 and OFES2 were small (0.084 $W/m^2$). However, there was heat advected
or diffused from the upper layer (at 300 m, the top face of the middle ocean layer). There was a difference of
approximately 0.747 $W/m^2$ in the VHD at a depth of 300 m in the Pacific Ocean and a difference of 0.701 $W/m^2$ in
the VHA. All these results led us to infer a VHD difference of 1.295 $W/m^2$ at a depth of 700 m in the Pacific Ocean,
with more heat diffusing downward in OFES1.
In the Atlantic Ocean, the average OHC trend, estimated by OFES1, was positive. It was, however, negative in
OFES2, with a difference of 0.129 $W/m^2$. A VHA of −1.585 $W/m^2$ was calculated for OFES2, which was 32% stronger
than that for OFES1. Additionally, more heat was lost through P1 into the Indian Ocean, and more heat was advected
into the Atlantic Ocean through the Drake Passage in the OFES1. Differences also existed in the heat advection
between the Atlantic Ocean and the Southern Ocean (P6) and the Arctic (P10) Oceans. The vertical heat transport
(VHA + VHD) at 300 m in the Atlantic Ocean (Tab. 2) was close between the two OFES data. The inferred VHD at
a depth of 700 m in the Atlantic Ocean was upward in both datasets, although it was stronger by 0.393 W/m$^2$ in the
OFES2.
The average OHC trend in the Indian Ocean was weakly negative for both OFES1 and OFES2. There was more heat
(by 0.142 W/m$^2$) advected downward at a depth of 700 m in the OFES2. Horizontally, 0.121 W/m$^2$ more heat was
acquired from the Atlantic Ocean (through P1) in the OFES1, however, there were only slight differences in the lateral
heat transport through the other passages connecting the Indian Ocean with other basins. The time-averaged VHD iat
700 m in the Indian Ocean was 0.423 W/m$^2$ in OFES1 and 1.083 W/m$^2$ in OFES2.
To summarize, the notable cooling trend in the Pacific and Atlantic Oceans (Fig.3), determined using OFES2 was
mainly generated from vertical heat transport (VHA + VHD) processes. For example, there was a net upward heat
advection at 300 m in the OFES2 Atlantic Ocean and a stronger downward heat advection at 700 m. As a result, more
heat was lost vertically in the middle layer of the OFES2 Atlantic Ocean compared to the OFES1 Atlantic Ocean.
**Table 3.** As for Tab. 2 but for the middle layer (300–700 m). VHA is at a depth of 700 m in this table.

| PACIFIC OCEAN (300–700 m) | | | | | | | | |
|---|---|---|---|---|---|---|---|---|
| | OHC | VHA | P3 | P4 | P5 | P7 | P8 | P9 | Residual |
| OFES1 | 0.017 | –0.096 | 1.208 | –0.026 | –1.056 | 0.044 | 0 | 0 | –1.679 |
| OFES2 | –0.034 | –0.084 | 1.247 | –0.030 | –0.917 | –0.040 | 0 | 0 | –0.384 |

| ATLANTIC OCEAN (300–700 m) | | | | | | |
|---|---|---|---|---|---|---|
| | OHC | VHA | P1 | P5 | P6 | P10 | Residual |
| OFES1 | 0.037 | –1.203 | –0.770 | 1.056 | 0.056 | –0.057 | 1.260 |
| OFES2 | –0.092 | –1.585 | –0.649 | 0.917 | 0.017 | –0.102 | 1.653 |

| INDIAN OCEAN (300–700 m) | | | | | | |
|---|---|---|---|---|---|---|
| | OHC | VHA | P1 | P2 | P3 | P7 | P8 | Residual |
| OFES1 | –0.010 | –0.519 | 0.770 | –0.043 | –1.208 | –0.044 | 0 | 0.423 |
| OFES2 | –0.013 | –0.661 | 0.649 | –0.043 | –1.247 | 0.040 | 0 | 1.083 |


*Lower layer*
OFES2 showed cooling in the bottom (700–2000 m) layer of each basin, but OFES1 showed overall warming (Tab.
4). In the Pacific Ocean, the VHA at 2000 m was downward and had a similar magnitude in the two OFES datasets.
Owing to the vertical coherence of the ACC, there was intense eastward heat advection through P3 and P5, even below
700 m, with the OFES2 showing higher advection. The horizontal heat advection through P4 and P7 was relatively
weak, and it was again larger in OFES2. For example, the MHA passing through P7 was more than one time larger in
the OFES2. In fact, more heat advected southward into the Indian Ocean through the ITF, which was found in all the
ocean layers (OFES1 showed a weak northward heat advection in the middle layer). As a result of these differences
and the estimated VHA and VHD at a depth of 700 m, we calculated a significant difference of approximately 1.252
W/m$^2$ in the VHD (in the downward direction) between the two OFES datasets at a depth of 2000 m in the Pacific
Ocean.

913 Unlike at 2000 m in the Pacific Ocean, OFES2 reflected that there was a significantly stronger downward heat

914 advection at 2000 m in the Atlantic Ocean. The dominant horizontal heat advections were through P1 and P5, with

915 the OFES2 showing stronger heat advection at both passages. We estimated a downward heat diffusion at a depth of

916 2000 m of 0.216 W/m$^2$ in the OFES1 Atlantic Ocean and an upward VHD of 0.383 W/m$^2$ in the OFES2 Atlantic

917 Ocean.

918 In the Indian Ocean, the calculated downward heat advection was twice as strong in the OFES1; there were also

919 some moderate differences in horizontal heat advection. The resulting VHD at 2000 m was upward in both OFES1

920 and OFES2, although it was much greater (by 0.455 W/m$^2$) in the latter.

921 In summary, the differences in the lateral heat advection through the major passages P1–P10 in the lower layer were

922 small, and the major drivers of the examined OHC differences between OFES1 and OFES2 were generated largely

923 from vertical heat transport (VHA + VHD), similar to the situation in the middle layer.

924 **Table 4.** As for Tab. 2 but for the lower layer (700–2000 m). VHA is at a depth of 2000 m.

| PACIFIC OCEAN (700–2000 m) | | | | | | | | |
|---|---|---|---|---|---|---|---|---|
| | OHC | VHA | P3 | P4 | P5 | P7 | P8 | P9 | Residual |
| **OFES1** | 0.058 | −0.126 | 0.951 | −0.047 | −1.120 | −0.035 | 0 | 0 | −1.341 |
| **OFES2** | −0.037 | −0.105 | 1.146 | −0.080 | −1.294 | −0.082 | 0 | 0 | −0.089 |

| ATLANTIC OCEAN (700–2000 m) | | | | | | |
|---|---|---|---|---|---|---|
| | OHC | VHA | P1 | P5 | P6 | P10 | Residual |
| **OFES1** | 0.014 | −0.029 | −0.974 | 1.120 | 0.066 | 0.105 | –0.216 |
| **OFES2** | −0.013 | −0.536 | −1.059 | 1.294 | 0.003 | −0.031 | 0.383 |

| INDIAN OCEAN (700–2000 m) | | | | | | | |
|---|---|---|---|---|---|---|---|
| | OHC | VHA | P1 | P2 | P3 | P7 | P8 | Residual |
| **OFES1** | 0.007 | −0.241 | 0.974 | −0.033 | −0.951 | 0.035 | 0 | 0.126 |
| **OFES2** | −0.018 | −0.120 | 1.059 | −0.052 | −1.146 | 0.082 | 0 | 0.581 |

925

## 4 Conclusions and Discussion

926

927 In this study, we estimated the OHC based on two eddy-resolution hindcast simulations, OFES1 and OFES2, with a

928 major focus on estimating their differences. The global observation-based dataset EN4 acted as a reference. The main

929 findings of this study are as follows:

930 1. Multi-decadal warming was clearly evinced in most of the global ocean (0–2000 m), especially in the EN4 and

931 OFES1 datasets. The warming was dominantly manifested as deepening of the neutral density surfaces (HV

932 component), with changes along the neutral surfaces (SP component) of regional importance.

933 2. Significant differences in the OHC (or potential temperature) were found between OFES1 and OFES2; the major

934 causes for these were fourfold. First, less surface heating or even cooling applied in OFES2. Second, the ITF was

almost one time stronger in the OFES2, especially in the top 300 m. Third, the differences in the intensity of the VHA
were large, particularly at the depth of 300 m in the Indian Ocean. Finally, remarkable differences in the vertical heat
diffusion were inferred.
3. Overall, the global and basin-integrated OHC estimates for the period 1960–2016 were reasonable for the top 700
m upon considering the OFES1 results. Below 700 m, multi-decadal climate changes derived from the OFES1 need
careful evaluations even though the estimates of global OHC between 700–2000 m are highly correlated with
observations. The notable differences between OFES2 and EN4 suggest that attention is clearly warranted while
concluding on multi-decadal climate changes based on OFES2.
Although we have detailed the OHC differences between the OFES1 and OFES2, and also analyzed the horizontal
and vertical heat transport in an attempt to understand the causes of these differences, further work is required for
improving this field. First, a direct calculation of the VHD is desirable to obtain a more reliable and accurate
comparison between the two OFES. In addition, decomposing the VHD into tidal mixing and mixed-layer vertical
mixing is also an interesting topic, and can help to isolate the effects of tidal mixing in the ocean state. We also expect
to see a detailed comparison of the wind stress from these two datasets over the 57 years. This is inspired by the work
of Kutsuwada et al. (2019) and our detection of the large differences in VHA. Considering the apparent differences in
the SP component among the three datasets, a comprehensive comparison of salinity between both OFES1 and OFES2,
along with observations, was required. This helped the community determine their choice of datasets for their research
purposes.
One may argue that the inability to spun-up completely could be the likely cause for the identified differences
between the OFES2 and other datasets since the OFES1 followed a 50-year climatological simulation but OFES did
not. However, large differences between the two OFES datasets can be seen in the temporal evolution of global and
basin OHCs, even during the last two decades. In addition, for example, **S2020** found that the Azores Current was
simulated in the OFES2 in the initial two decades, however, it disappeared after 1970. These, to some extent, weaken
the spin-up argument, although it does not rule out the possibility completely. OFES2 was not expected to be highly
sensitive to the spin-up issue because the starting conditions are from OFES1. There were indeed some improvements
in the OFES2 during the recent decades, for example, from to 2005–2016 (not shown here). Two potential explanations
are as follows: First, the model was well spun-up after a couple of decades of integration; second, improvements in
the reanalysis of atmospheric forcing data contributed to improvements in simulation.
As mentioned above, results based on EN4 should not be considered as the *truth*. Several factors such as mapping
methods and data assimilated impact the resulting quality of the observational-based product, and might consequently
alter our conclusions. As a preliminary test of robustness, we compared the temporal evolution of the OHC (Fig. S10)
and the spatial patterns of the long-term potential temperature trend (Fig. S11) determined using EN4 and two datasets,
G10 and IAP. G10 is the most up-to-date version of EN4 datasets (EN4.2.2) with bias-corrected following Gouretski
and Reseghetti (2010) and IAP is the dataset from the Institute of Atmospheric Physics (Cheng and Zhu, 2016). The
primary difference between EN4 (bias-corrected following Levitus et al. (2009)) and G10 lies in the bias correction
methods, whereas IAP differs from EN4 in assimilated datasets, mapping methods, and among others. The large
similarities between EN4 and G10 suggest that the different correction methods do not lead to notable differences in
the resulting state estimates. On the other hand, there were some differences between the IAP and both EN4 and G10.
This may indicate that the applied mapping method causes some discrepancies among different oceanic products,
which is consistent with Cheng and Zhu (2016). Nonetheless, this preliminary test shows that our primary conclusions
are unlikely to be altered when choosing different observational-based datasets for comparisons.
Finally, in absence of any observation-based constraints, the OFES products, especially the OFES1, have captured
some of the warming and cooling trends shown by EN4 and in the literature. However, clear differences between the
two OFES datasets and EN4 suggest the importance of observational data in improving the performance of a hindcast
simulation. The significant differences in the vertical heat diffusion between the two OFES datasets also suggest that
special attention should be given to the validation of the vertical mixing scheme in future ocean modeling.

**Author contributions:** F.L conceived the study. All authors contributed to the details of study design. F.L conducted
the calculations and analysis. F.L drafted the manuscript; Z.L and X.H.W improved the writing.

**Competing interests:** The authors declare that they have no conflict of interest.

**Acknowledgements:** This is publication No. 87 of the Sino-Australian Research Consortium for Coastal Management
(previously the Sino-Australian Research Centre for Coastal Management). This work was supported by the Key
Special Project for Introduced Talents Team of the Southern Marine Science and Engineering Guangdong Laboratory
(Guangzhou; GML2019ZD0210). The authors thank Dr. Peter McIntyre for improving the manuscript. The authors
acknowledge public access to the data used in this paper from the UK Meteorological Office and the JAMSTEC.
Constructive comments from the editor and two anonymous reviewers greatly improved the manuscript.

**Code and data availability:** OFES1 and OFES2 are based on the MOM3, available at https://github.com/mom-
ocean/MOM3. Code for decomposing the potential temperature: http://www.teos-10.org/software.htm. Original EN4
data: https://www.metoffice.gov.uk/hadobs/en4/download-en4-2-1.html. Original OFES1 temperature and salinity
data: http://apdrc.soest.hawaii.edu/dods/public_ofes/OfES/ncep_0.1_global_mmean. Due to a data security incident,
access to the OFES2 data has been temporarily suspended. The data and codes (including the publically available
scripts for completion) needed to reproduce the results of this paper are archived on Zenodo
(https://doi.org/10.5281/zenodo.5205444). The archived data are annual mean values calculated from the original data.

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
