# Peer review of "Comparison of ocean heat content estimated using two eddy-resolving hindcast simulations based on OFES1 and OFES2"

_Geoscientific Model Development, 2021_

## Author Comment (AC2)

Dr. Qiang Wang

Topic editor

Geoscientific Model Development

**Re: gmd-2021-95**

Dear Dr. Qiang Wang and reviewer

Thank you for handling and reviewing our manuscript entitled "Comparison of ocean heat content from two eddy-resolving hindcast simulations with OFES1 and OFES2" to be considered for publication in the GMD. We appreciate your very constructive comments and have addressed accordingly.

Major comments:

1. *It is interesting that the ocean heat content changes primarily by the change in isopycnal depth. Does the total heat content calculation depend upon the calculation of heat content change by heaving motions? It would be good to provide corroborating evidence regarding the heat content decomposition, such as an independent calculation of total heat content variability.*

***Response:*** In this paper, we decomposed the potential temperature change into heave and spiciness components as a method to identify the way how the water warms or cools. In Figs. 2-6 and 8, we have OHC, HV (heave) and SP (spiciness), and please be noted that the OHC here is calculated directly from the potential temperature following Eq. 1 (Line#131 in the clean version). To make it clearer, we specify in the revised manuscript that "The OHC hereafter is directly calculated from the potential temperature". (Line 209-210 in the clean version)

2. *Abstract: Heat transport is stated to not always be responsible for ocean heat content changes. Doesn't it have to be either the heat transport or the air-sea flux, given thermodynamic energy conservation? Heat content storage will be the residual of these terms. It is puzzling to consider where the thermodynamic energy is transported. The manuscript would benefit with a closed energy budget analysis, which may require the deep ocean and/or the Arctic to accurately assess where the energy goes.*

***Response:*** In the first version of this manuscript, we presented the pattern of net surface heat flux, horizontal and vertical heat transport. Their pattern is geographically similar, although differences are also clear in some places. However, as it is not easy to link this pattern to the examined OHC differences between the two OFES products, we therefore speculate that the OHC differences may result from the discrepancies in the mixing, especially the vertical mixing, given the OFES1 and OFES2 used KPP and mixed layer vertical mixing, respectively. Although the spatial pattern helps to qualitatively analyse the differences, a detailed heat budget is desirable indeed. In the revised manuscript, we used the currently available data to calculate the inter-basin heat exchange and vertical advection of heat, as can be found in Tabs. 4-5. Caused by a temporal suspension of data from JAMSTC, we are not able to access the vertical diffusivity data of the OFES2 (OFES1 itself does not provide the vertical diffusivity data). Alternatively, we approximately took the residual of the OHC variations and the heat input (net surface heat, inter-basin heat exchange and basin-integrated vertical heat advection at a given depth (500m and 1500 m in this paper) as the vertical diffusion of heat. This indirect method may suffer from some errors, but could help to identify the major vertical mixing distinctions between the OFES1 and OFES2. As the new results show, we found there is less heating for the major basins in the OFES2. The horizontal heat advection is largely similar but the OFES2 has a much stronger meridional heat transport associated with the Indonesian Throughflow (ITF). It was also found that the vertical heat advection differs significantly. Therefore, it was claimed that the marked OHC differences may arise from the less heat

input from the atmosphere, significantly different vertical heat advection and the inferred vertical heat diffusion.

Minor comments:

*1. L29: does this sentence equate objective analysis and ocean reanalysis? They are normally considered to be distinct*

**Response:** Yes, these two are different. In this sentence, we want to say that different types of oceanic data are available for the 4D studies of the ocean thermal state, e.g. objective analysis (e.g. EN4) and ocean reanalysis (e.g. ECMWF ORAS5). To make it clearer, this sentence was revised to "Typical examples of these approaches include the objective analysis of observational data **and** ocean reanalysis by physical ocean models constrained by observations". (Lines# 37-38 in the clean version)

*2. How are ocean heat content changes related to algorithmic changes between OFES 1 and 2?*

**Response:** These two use the same model MOM3, same horizontal resolution and horizontal mixing scheme. That's is, the core of the algorithmic should be the same between these two. However, they used different surface forcing, vertical mixing schemes, different initial conditions. As no vertical diffusion coefficient is available, it could be difficult to evaluate these impacts.

*3. How did the authors justify restricting their analysis to the upper 1400 meters? Their rationale following Emery (2001) and Wunsch (2011) is not compelling. Did these previous works suggest that ocean disequilibrium occurs suddenly at 1400 meters depth? (That would be surprising.) During the time period of interest, i.e., 1950-present, why should the deep ocean be in equilibrium?*

**Response:** The primary reasons we consider to focus only on the upper 1500 m (There was a misinterpretation of water depth in the previous version, it should be 1500 m rather than 1400 m, which has been rectified in the new revised manuscript) are mainly two-fold. 1) we analyse the OHC variations largely from the perspective of water masses. As defined in Emery (2001), the world ocean was divided into three layers (0-500 m, 500-1500 m and below 1500 m). 2) the observational data ingested by the EN4 is largely confined to the upper ocean, with much lower density of data in the deep and abyssal ocean.

Furthermore, the ingested data in the EN4 version we used here is bias-corrected following Levitus et al. (2009), in which only the upper ocean is considered. Another less important reason is that the maximum depth among these three data is significantly different.

*4. For comparison purposes, does the NCEP reanalysis give air-sea conditions every 6 hours, as opposed to 3 hours in OFES2?*

*Response:* The OFES1 was forced by the daily NCEP forcing, as can be found in (Line# 88 in the clean version).

*5. L86: "Validation" is not possible with EN4 as it is also an incomplete and uncertain product.*
*Response:* This was changed to "objectively evaluate". (Line#102 in the clean version)

*6. Figure 1: What happens in the Arctic? What error is incurred by eliminating the Arctic?*
*Response:* It is very important to look into the state of the Arctic. However, the OFES1 (OFES2) is confined to 75°S-75°N (76°S – 76°N). Also, the potential temperature decomposition into heave and spice is valid between 80°S and 64°N. In addition, a sea-ice model applies in the OFES2 but not in the OFES1. We therefore, did not include the Arctic in this work.

*7. Figure 2 is fascinating, if correct. What is going on with OFES2?*
*Response:* As the Figs. 12-14 shows, the large-scale pattern of the surface heat flux and the horizontal and vertical heat advection is much the same. But there is generally less heat input or more heat loss in the OFES2. In addition, from the calculated heat flux through the major water passages around the world ocean and basin-integrated vertical heat flux at 500 m and 1500 m, we can see that the horizontal heat flux is close to each other, except the one associated with the Indonesian Throughflow, for which the OFES2 shows much stronger heat flux. There exists large differences in the vertical heat flux between the OFES1 and OFES2, and we also infer that the vertical heat diffusion is largely different between the two data.

*8. L161: dividing by 56 "years".*
*Response:* This was addressed (Line# 222 in the clean version).

*9. What does "SP" stand for?*
*Response:* The SP means the spiceness component of the potential temperature change and similarly the HV stands for the heaving. (Line#58 in the clean version)

*10. L242: 10 to the 6th power*
*Response:* This was addressed (Line#318 in the clean version).

*11. Does OFES 2 fit surface data (i.e., SST)?*
*Response:* The OFES2 developer examined the SST comparison between the OFES2 and WOA13 over 2005-2012 and shows good results.

*12. Table 4 doesn't seem very useful with the inexact metrics for the water-mass source properties.*

**Response:** Tabs. 4-5 are built by following the definition of Emery (2001), to help the readers to have a first impression on the water masses, should they are not familiar with these. To be more concise, we move these two tables into supplementary section. Alternatively, we roughly label the geographic coverage of the major water masses in Figs 6 and 8.

---

## Author Response (AR1)

Dr. Qiang Wang

Topic editor

Geoscientific Model Development

**Re: gmd-2021-95**

Dear Dr. Qiang Wang, Dr. Juan A. Añel and reviewers

Thank you for handling and reviewing our manuscript entitled "Comparison of ocean heat content from two eddy-resolving hindcast simulations with OFES1 and OFES2" to be considered for publication in the GMD. We appreciate your very constructive comments and have addressed accordingly.

**I. Response to Dr. Juan A. Añel, the Executive editor**

Major comments:

1. *We have checked your manuscript, and unfortunately, at the moment, it does not comply with our 'Code and Data Policy'. You compare results with OFES1 and OFES2; however, nothing is said about how to access models. To state that they are based on MOM3 is not enough. Moreover, Github, as we state in our policy and Github itself on its website, it is not a suitable repository for long-term archival. Also, personal repositories in institutional web pages are not valid. Therefore, please, provide clear indications about how to access the OFES1 and OFES2 models, complying with our policy, and move your code and scripts to one of the suitable repositories that we list before the end of the Discussions period and make the necessary changes in the manuscript in potential reviewed versions.*

*Response:* Our work mainly evaluates the two eddy-resolving hindcast simulations conducted by the JAMSTEC. This work is motivated by the fact that the multi-decadal high-resolution model data is valuable for both the oceanography and climate communities. Therefore, we felt it is both incentive and interesting to have a comparison of the ocean heat content over a long period, which is expected to provide important references for the future OFES2 users, as the OFES1 has been widely used in different fields, including the OHC.

We download the original monthly data from the official web of each product and save it into annual mean files. At presents, people can still freely access to the original EN4 and OFES1 data, but the OFES2 is still subject to a temporal suspension due to data security incident (http://www.jamstec.go.jp/e/about/informations/notification_2021_maintenance.html). As an alternative, we have put all the data and scripts into a package and archived it in Zenodo as you kindly suggested. Therefore, we made some modifications in the section of code and data avaiailability as follows:

**Code and data availability:** OFES1 and OFES2 are based on the MOM3, available at https://github.com/mom-ocean/MOM3. Code for decomposing the potential temperature: http://www.teos-10.org/software.htm. Original EN4 data: https://www.metoffice.gov.uk/hadobs/en4/download-en4-2-1.html. Original OFES1 temperature and salinity data: http://apdrc.soest.hawaii.edu/dods/public_ofes/OfES/ncep_0.1_global_mmean. Due to a data security incident, access to the OFES2 data has been temporarily suspended. The data and codes (including the publically available scripts for completion) needed to reproduce the results of this paper are archived on Zenodo (https://doi.org/10.5281/zenodo.5205444). The archived data are annual mean values calculated from the original data. (Lines#940−946 in the clean version).

2. *I am aware of what you mention. Being bad enough that the research is based on a model that we can not audit, the text should include an explicit mention of this. At the moment, the fact that the authors have not run the model but only tested downloaded data is not clear enough. Also, my comment applies to the Teos-10 software. Here there are two issues: First, the authors do not clarify what version of this software they use: Fortran, C, etc. Also, the software is on a webpage that we cannot consider a trustful repository. According to the license of Teos-10, the code can be redistributed. Therefore, the authors should upload the code they have used to one of the repositories that we can accept and provide the DOI for it. Indeed, currently, the 'Excel' version seems to be already stored in Zenodo: https://zenodo.org/record/4751051. Moreover, a Github repository exists. Perhaps it can be used to fork the code and upload the version of the manuscript to Zenodo.*

*https://github.com/TEOS-10. Another issue that I forgot to mention in my previous comment, the link to MOM3 in the 'Code and Data Availability' section is broken. At the moment, it points to "https://github.com/mom-460" instead of " https://github.com/mom-ocean/MOM3."*

*Finally, given the small size of some archives (as those of EN4.2.1), it would be good to curate them in Zenodo if possible. For example, today, the JAMSTEC servers are offline because of a security breach. Hopefully, they will come alive at some point, but at the moment, it is not possible to access part of the data for this manuscript.*

**Response:** In the revised version, we made it clear that two OFES simulations were conducted by the JAMSTEC (Lines#44–48 in the clean version). We specify the code version we used in this work (Line#178 in the clean version). All the data and codes necessary to reproduce the results are archived in Zenodo (https://doi.org/10.5281/zenodo.5205444). We double checked the link https://github.com/mom-ocean/MOM3 and it works now.

**II.     Response to reviewer#1**

Major comments:

1. *It is interesting that the ocean heat content changes primarily by the change in isopycnal depth. Does the total heat content calculation depend upon the calculation of heat content change by heaving motions? It would be good to provide corroborating evidence regarding the heat content decomposition, such as an independent calculation of total heat content variability.*

***Response:*** In this paper, we decomposed the potential temperature change into heave and spiciness components as a method to identify the way how the water warms or cools. To make it clearer, we specify in the revised manuscript that "The OHC hereafter is directly calculated from the potential temperature". (Lines#215−216 in the clean version).

In addition, by taking the EN4 as an example, we also directly compared the OHC derived from the potential temperature and the sum of HV and SP in the following Fig. 1. It clearly shows that there is a good correspondence between the OHC and the sum of HV and SP, with a relatively small residual.

[Figure]

**Figure 1.** A comparison between the OHC (black solid line) and the sum (red solid line) of HV and SP derived from the EN4 data. The blue dash line is the residual (OHC − HV − SP). From the left to the right, it is the global ocean (GLO), the Pacific Ocean (PAC), the Atlantic Ocean (ATL) and the Indian Ocean (IND). The top row is for 0-300 m, the middle row for 300− 700 m and the bottom row for 700-2000 m.

2. *Abstract: Heat transport is stated to not always be responsible for ocean heat content changes. Doesn't it have to be either the heat transport or the air-sea flux, given thermodynamic energy conservation? Heat content storage will be the residual of these terms. It is puzzling to consider where the thermodynamic energy is transported. The manuscript would benefit with a closed energy budget analysis, which may require the deep ocean and/or the Arctic to accurately assess where the energy goes.*

***Response:*** In the first version of this manuscript, we presented the spatial pattern of net surface heat flux, horizontal and vertical heat transport. Their pattern is geographically similar, although differences

are also clear in some places. However, as it is not easy to link this pattern to the examined OHC differences between the two OFES products, we therefore speculate that the OHC differences may result from the discrepancies in the mixing, especially the vertical mixing, given the OFES1 and OFES2 used KPP and mixed layer vertical mixing (with considerations of tidal mixing), respectively. Although the spatial pattern helps to qualitatively analyse the differences, a detailed heat budget is desirable indeed. In the revised manuscript, we followed you very constructive suggestions and used the currently available data to calculate the inter-basin heat exchange and vertical advection of heat, as can be found in Tabs. 2–4. Caused by a temporal suspension of data from JAMSTC, we are not able to access the vertical diffusivity data of the OFES2 (OFES1 does not provide the vertical diffusivity data). Alternatively, we approximately took the residual of the OHC variations and the heat input (net surface heat, inter-basin heat exchange and basin-integrated vertical heat advection at a given depth (300, 700 and 2000 m in this paper) as the vertical diffusion of heat. This indirect method may suffer from some errors, but could help to identify the major vertical mixing distinctions between the OFES1 and OFES2. As the new results show, we found there is less surface heating for the major basins in the OFES2. The horizontal heat advection is largely similar but the OFES2 has a much stronger meridional heat transport associated with the Indonesian Throughflow (ITF). This enhancement of ITF is related to the applied internal tidal mixing (Sasaki et al. 2018). It was also found that the regional vertical heat advection may differ significantly, for example, at the depth of 300 m in the Indian Ocean (Tab. 2). Therefore, it was claimed that the marked OHC differences may arise from the less heat input from the atmosphere, significantly different vertical heat advection and the inferred vertical heat diffusion.

Minor comments:

*1. L29: does this sentence equate objective analysis and ocean reanalysis? They are normally considered to be distinct*
**Response:** Yes, these two are different. In this sentence, we want to say that different types of oceanic data are available for the 4D studies of the ocean thermal state, e.g. objective analysis (e.g. EN4) and ocean reanalysis (e.g. ECMWF ORAS5). To make it clearer, this sentence was revised to "These approaches include the objective analysis of observational data and ocean reanalysis combing physical ocean models with observations". (Lines# 38–39 in the clean version)

*2. How are ocean heat content changes related to algorithmic changes between OFES 1 and 2?*
**Response:** These two use the same model MOM3, same horizontal resolution and horizontal mixing scheme. That's is, the core of the algorithmic should be the same between these two. However, they used different surface forcing, vertical mixing schemes, different initial conditions. As no vertical diffusion coefficient is available, it could be difficult to directly evaluate these impacts. However, as shown in Tabs. 2-4, we inferred that there are profound disparities in the vertical heat mixing as a result of different mixing scheme.

*3. How did the authors justify restricting their analysis to the upper 1400 meters? Their rationale following Emery (2001) and Wunsch (2011) is not compelling. Did these previous works suggest that ocean disequilibrium occurs suddenly at 1400 meters depth? (That would be surprising.) During the time period of interest, i.e., 1950-present, why should the deep ocean be in equilibrium?*

**Response:** The primary reasons we consider to focus only on the upper 1500 m in the original version (There was a misinterpretation of water depth in the previous version, it should be 1500 m rather than 1400 m, which has been rectified in the new revised manuscript) are mainly two-fold. 1) we analyse the OHC variations largely from the perspective of water masses. As defined in Emery (2001), the world ocean was divided into three layers (0-500 m, 500-1500 m and below 1500 m). 2) the observational data ingested by the EN4 is largely confined to the upper ocean, with much lower density of data in the deep and abyssal ocean. In fact, the vast of available observations is confine to the upper 700m over the last 50+ years, as stated in (Hakkinen et al. 2016). As we want to take the EN4 as a reference, we thought it might be safer to focus on the depth range where more observational data was ingested.

Furthermore, the ingested data in the EN4 version we used here is bias-corrected following Levitus et al. (2009), in which only the upper ocean is considered. Therefore, for instance, the XBT profiles below 700m will be corrected using the correction values provided for 700 m (personal communication with the Met Office Hadley Centre). Another less important reason is that the maximum depth among these three data is significantly different, that is, a comparison of the full-depth OHC may not be justified. A full justification can be found in the clean version (Lines#122–134).

That said, in this new version, we have three vertical layers: 0–300 m, 300–700 m and 700–2000 m. This, on the one hand, follows the conventional vertical division of the ocean (many previous studies considered the ocean between 0–700 m and 0–2000 m or between 700–2000m). However, we found that above 300 m, the OFES2 generally has a better performance than the OFES1 when comparing to the EN4. We therefore, feel it is necessary to have a near-surface layer (0–300 m).

*4. For comparison purposes, does the NCEP reanalysis give air-sea conditions every 6 hours, as opposed to 3 hours in OFES2?*

**Response:** The OFES1 was forced by the daily NCEP forcing, as can be found in (Line# 86 in the clean version).

*5. L86: "Validation" is not possible with EN4 as it is also an incomplete and uncertain product.*

**Response:** This was changed to "To evaluate the OHC objectively from the two OFES data". (Line#99 in the clean version)

*6. Figure 1: What happens in the Arctic? What error is incurred by eliminating the Arctic?*

**Response:** It is very important to look into the state of the Arctic. However, the OFES1 (OFES2) is confined to 75°S-75°N (76°S – 76°N). Also, the potential temperature decomposition into heave and spice is valid only between 80°S and 64°N. In addition, a sea-ice model applies in the OFES2 but not in the OFES1. We therefore, could not include the Arctic in this work.

*7. Figure 2 is fascinating, if correct. What is going on with OFES2?*

**Response:** The original Fig. 2 shows the comparison of the time evolution of OHC, HV and SP between the three data. The OHC was directly calculated from the potential temperature data. Although this figure is now removed, we found that discrepancy of OHC evolution between the OFES2 and EN4 is remarkable for the ocean below 700 m. Based on the heat budget analysis for each major basin, we found that the surface heat flux, enhanced Indonesian Throughflow, vertical heat advection and the inferred vertical heat diffusion are the major causes of these notable differences.

*8. L161: dividing by 56 "years".*

**Response:** Yes, this should be 56 years. In this new version, we remove the original Tab. 2, as we have shown the rolling trend in Figs. 5-7.

*9. What does "SP" stand for?*

**Response:** The SP means the spice component of the potential temperature change and similarly the HV stands for the heave. (LineS#164–165 in the clean version)

*10. L242: 10 to the 6th power*

**Response:** This was addressed (Line#143 in the clean version).

*11. Does OFES 2 fit surface data (i.e., SST)?*

**Response:** The OFES2 developer examined the SST comparison between the OFES2 and WOA13 over 2005-2012 and shows good results (Sasaki et al., 2020). This seems to be consistent with our new finding that the OFES2 has a better performance in the top 300 m.

*12. Table 4 doesn't seem very useful with the inexact metrics for the water-mass source properties.*

**Response:** Tabs. 4-5 are built by following the definition of Emery (2001), to help the readers to have a first impression on the water masses, should they be not familiar with these. In this new version, we removed this table and no longer used the original water-mass definition. Instead, we analyzed the water mass in the density-coordinate, which was also a response to the reviewer#2.

**III. Response to reviewer#2**

Major comments

*1.	The decomposition into HV and SP. One major conclusion of this paper is "There was an OHC increase in most of the global ocean over a 57-year period, mainly a result of vertical displacements of neutral density surfaces.". However, I don't think it is a robust conclusion given the fact that neither OFES1 nor OFES2 well simulate the OHC changes globally or at each major ocean basin.*

**Response:** In the original version, we failed to make it clear that this conclusion is specific for multi-decadal scale and mainly in the EN4 and OFES1. As shown in Figs. 2–4 and also 14–16, the majority of the global ocean has an overall warming trend. Indeed, the OFES2 largely shows a cooling trend, especially for the water below 300 m. In the new version, we changed it into "OHC increased in most of the global ocean above 2000 m in the EN4 and OFES1 over 1960–2016, mainly a result of deepening of neutral density surfaces, with variations along the neutral density surfaces of regional importance." (Lines#12–14)

*2.	The investigation of heat flux and heat transport are not well designed and not useful. To examine the mechanisms for the change of OHC, you have to check the trends in heat flux and heat transport, not the climatology field (Figs. 10, 11). In another word, you have to know where more heats are input into the ocean and how they are transported.*

**Response:** In the original version, we did not make it very clearly why we presented the time-averaged heat flux and heat transport patterns. As shown in the original zonal-averaged OHC distribution (Figs. 4-5) and the spatial pattern of the potential temperature change (Figs. 6-7), we are focusing the differences between the mean of the last three years (2014-2016) and the beginning three years (1960-1962). A time-averaged field over this period is the ratio of the accumulative heat flux or heat transport over the total time length. Therefore, it (if multiplied with time) can be related to the total OHC change over the whole period. To be more quantitative, we calculated the basin-wide heat flux, inter-basin heat exchange and vertical heat advection, and we also inferred the vertical heat diffusion in the new version. It is now clearer that there is generally less surface heating entering into the three major basins in the OFES2. The horizontal heat advection through most of the inter-basin passages are much the same, but the Indonesian Throughflow (ITF) is around two time stronger in the OFES2 (this can partially explain the differences in the Pacific and Indian Ocean). The regional vertical heat advection and diffusion can be also significantly different between the OFES1 and OFES2.

Indeed, a temporal evolution of surface heat flux and heat transport can make things clearer. Therefore, we plotted such figures and put them in the section of support information (Figs. S7-9).

*3.	The water mass analyses in section 3.3 are also problematic, because the water masses are defined by the density or the temperature/salinity range as in Tables 4,5, however, the figures 6 and 7 are presented at z-coordinate, so the discussions are very confusing and not corresponding to the plot.*

**Response:** In the original version, we adopted the definition of water mass from (Emery 2001). That's why have two vertical layers (0–500 m and 500–1500 m). The Figs. 6–7 were analysed by following the geographic locations of these different water masses (Figs. 3–4 in Emery 2001). But yes, we agree that it is better and clearer to analyse the water mass in the density-coordinate. Therefore, in the new

version, by following your constructive suggestions, we added two new figures (Figs. 17–18) where we presented the HV and SP in the neutral density coordinate. This is similar to (Hakkinen et al. 2016).

4.  *Section 3.2. The zonal integrated OHC, HV and SP. This section superficially described the results without any in-depth analyses or insights. It is not useful for the audience.*
***Response:*** We agree that the original analysis in the original section 3.2 is preliminary and loose. In the new version, we calculated the rolling trend of the zonal integrated OHC, HV and SP. We combine the potential temperature change with its HV and SP component and present a more detailed discussion. For example, we discussed when and where these three datasets are similar to or different from each other, and whether this similarity or disparity is related to their HV or SP.

5.  *Section 3.1. Why not also provide the global or basin time series for surface heat flux for comparison? Globally, the heat content change is balanced by surface heat exchange. The decomposing into HV and SP does not help to understand the mechanisms here.*
***Response:*** We agree that a comparison of surface heat flux is essential. In this new version, a comparison of surface heat flux (basin-wide and time-averaged) is shown in Tabs. 2-4. We also presented the time evolution of basin-wide surface heat flux for the Pacific, Atlantic and Indian Ocean in the support information.

  Yes, the HV and SP can help to understand what the dominant ways for the potential temperature change (by the vertical deepening of the neutral density surface of along the neutral density surface and therefore salinity change involved) are, but not be able to well explain the mechanisms.

6.  *I expect an answer of why is OFES2 so different from OFES1, so a formal ocean heat budget analysis should be done.*
***Response:*** We agree and understand that a complete heat budget analysis is essential to answer the major reasons of the remarkable differences between the two OFES data. In this new version, we tried to conduct a reasonable heat budget analysis with all the available data we have (please be noted that the vertical diffusivity is not available from the OFES1; although the OFES2 output the vertical diffusivity, the data service is temporarily suspended due to a security incident). More specifically, as shown in Tabs. 2–4, we compared the surface heat flux, vertical heat advection, inter-basin heat exchange and we inferred the vertical heat diffusion. It is found that there is generally less surface heat entering the major basin in the OFES2, the vertical heat advection can be significantly different (e.g., at the 300 m depth of the Indian Ocean), much stronger ITF is simulated in the OFES2. Moreover, the inferred vertical heat diffusion is also profoundly different, which is due to their different vertical mixing scheme and the inclusion of internal tidal mixing in the OFES2.

7.  *Another conclusion "However, these differences, more specifically in the heat transport, were only partially responsible for the OHC differences." Is not tenable, because I did not see an analysis for ocean heat budget, and the current analyses are wrong because only climatological heat flux and transport are shown.*
***Response:*** Please see our reply to the comment #2 and #6.

*8.     Final conclusion in the abstract "The marked OHC differences may arise from the different vertical mixing schemes and may impact the largescale pressure field, and thus the geostrophic current". This is a full speculation without any evidence.*

***Response:*** Yes, we agree that we should have shown the differences of vertical diffusion before making this speculation. To be more convincing, we calculated and compared the vertical heat diffusion between the OFES1 and OFES2, and it confirms that there are large discrepancies of vertical heat diffusion in these two OFES data. But, please be noted that we inferred these vertical heat diffusions in an indirect way, as we have no access to the vertical diffusivity at the time of this writing.

*9.     Is the decomposing of EN4 data into HV and SP consistent with previous results? How large is the uncertainty behind the decomposing method given the data errors?*

***Response:*** Our decomposition of the EN4 data into HV and SP has a good correspondence to a previous study by (Hakkinen et al. 2016), our Fig. 17 and their Fig. 6, despite the time period is slightly different and they are based on a previous version of EN4 product. Also, we plotted the residual of OHC and the sum of HV and SP for all the major basins and all the three depth ranges by using the EN4 data, as shown in the Fig. 1 of this document. It clearly shows that the residual is small in general. Therefore, our decomposition is reliable. Please be noted that a small residual can be hardly avoided, due to the vertical interpolation, air–sea interactions and large vertical temperature (Desbruyères et al. 2017; Hakkinen et al. 2016).

*10.     For zonal integrated OHC analyses, why some regions models are closer to observations and some places are not? What are the possible reasons and what are the implications? Again, an ocean heat budget analysis at each zonal band might help to identify if the difference comes from surface or ocean heat transport.*

***Response:*** In this new version, we have conducted a basin-wide heat budget analysis to elaborate the causes of the differences between the OFES1 and OFES2. A possible explanation for the latitudinal dependence is that there are differences in the wind stress forcing in these two simulations but a larger thermocline (which is latitudinal dependent) responses to the wind stress differently (Kutsuwada et al. 2019). We expect a more detailed exploration of the model differences, but this is beyond of our current scope.

*11.     Section 3.4: diving the ocean by 0-500m and 500-1400m will cross-cut several different water masses. It is really strange to use vertical levels in water mass analyses.*

***Response:*** Please see our response to the comment # 3.

**References:**

Desbruyères, D., E. L. McDonagh, B. A. King, and V. Thierry, 2017: Global and Full-Depth Ocean Temperature Trends during the Early Twenty-First Century from Argo and Repeat Hydrography. *Journal of Climate*, **30,** 1985-1997.

Emery, W., 2001: Water Types and Water Masses. *Encyclopedia of Ocean Sciences*, **4,** 3179-3187.

Hakkinen, S., P. B. Rhines, and D. L. J. J. o. C. Worthen, 2016: Warming of the Global Ocean: Spatial Structure and Water-Mass Trends, **29,** 4949-4963.

Kutsuwada, K., A. Kakiuchi, Y. Sasai, H. Sasaki, K. Uehara, and R. Tajima, 2019: Wind-driven North Pacific Tropical Gyre using high-resolution simulation outputs. *Journal of Oceanography*, **75,** 81-93.

Sasaki, H., S. Kida, R. Furue, M. Nonaka, and Y. Masumoto, 2018: An Increase of the Indonesian Throughflow by Internal Tidal Mixing in a High-Resolution Quasi-Global Ocean Simulation. *Geophysical Research Letters*, **45,** 8416-8424.

---

## Author Response (AR2)

Dr. Qiang Wang

Topic editor

Geoscientific Model Development

**Re: gmd-2021-95**

Dear Dr. Qiang Wang and reviewer,

Thank you for handling and reviewing our manuscript entitled "Comparison of ocean heat content from two eddy-resolving hindcast simulations with OFES1 and OFES2" to be considered for publication in the GMD. We appreciate your very constructive comments and have addressed accordingly.

**I.    Response to reviewer#2**

Minor comments

*1. The uncertainty in observational dataset should be tested and bring into serious attention. It is well-known the observation is not perfect, it will add a non-negligible error when comparing with models. I would rather suggest to include 1-2 other data products to test the robustness of the results, I don't think you need to replicate all results, just test several your key results will be sufficient.*

***Response:*** In the new revised version, we added another two observational-based datasets. The first one is the most up-to-date EN4 version, EN4.2.2 with bias corrected following Gouretski and Reseghetti (2010), we called it G10; and the second is the temperature dataset from the Institute of Atmospheric Physics (Cheng and Zhu, 2016), we called it IAP. We calculated and compared the temporal revolution (Fig. S10) and spatial trend distribution (Fig. S11) between EN4 and G10 and IAP. Overall, these three datasets show high similarities but certain differences can be identified, especially between IAP and other two. This is consistent with previous studies that the mapping method may produce large differences in the state estimate. A discussion can be found li lines#929–939 of the clean version of the revised manuscript.

*2. Maybe I missed something, but how the uncertainty range in the new figures (Fig.5, 6 for example) is estimated?*

***Response:*** The linear trend was calculated using the multiple linear regression using least squares, and we used the 95% confidence level (lines# 190–192 of the clean version of the revised manuscript).

*3. Line 156-157: which version of EN4, it gives several options (corresponds to different bias correction methods).*

***Response:*** We used the EN4.2.1, with bias corrected following Levitus et al (2009). Lines# 100–101 of the clean version of the revised manuscript.

*4. Fig.8-11 and other figures and section 3.2. I'm wondering why 10-year rolling trends were used. It seems not be able to remove ENSO. I would suggest using 15-year rolling if you want to see the low-frequency signals.*

***Response:*** We did not make it clear enough. The rolling trend was used to compare these three datasets in each 10-year window with rolling method, in order to see whether any improvements of the two OFES datasets with time. This was not intended to focus on the low-frequency signals. Lines# 310–311 of the clean version of the revised manuscript.

---

## Author Response (AR3)

Dr. Qiang Wang

Topic editor

Geoscientific Model Development

**Re: gmd-2021-95**

Dear Dr. Qiang Wang

Thank you for handling and reviewing our manuscript entitled "Comparison of ocean heat content from two eddy-resolving hindcast simulations with OFES1 and OFES2" to be considered for publication in the GMD. We appreciate your very constructive comments and have addressed accordingly.

Minor comments

*1. 1) Starting from line 929, "One reviewer raised the concern on the uncertainty in the observational datasets (EN4) and suggested to add one or two more observation-based datasets to reproduce some of our results here."*
*This should not be the motivation to check different datasets. The reason is that the data quality can influence your findings. The current text does not read like a paper, rather like a reply letter to reviewers. Please adjust this paragraph, and also pay attention to the choice of location for adding this paragraph. The discussion should end with a clear message.*
**Response:** In the new revised version, we changed the first couple of sentences of this added paragraph as "*As mentioned above, the EN4 should not be taken as the truth. Factors such as mapping methods and data ingested impact the resulting quality of those objective-analysis products and may alter our conclusions here consequently. As a preliminary test of robustness, we compared ...*". (Lines#963–965). This paragraph ends with "*Nonetheless, this preliminary test shows that our primary conclusions are unlikely to be altered when choosing different observational-based datasets for comparisons*". (Lines#974–975).

As for the location of this paragraph, we kept it unchanged after careful comparisons and thinkings. The reason for keeping unchanged is that this paragraph is an additional (further) discussion that is not directly related to the main body of this manuscript, different from the prior discussions in this sense. The last paragraph is more as a reminding message to both the users and model developers.

*2. In both the abstract and conclusion section, readers expect to see a clear message about the improvement (if any) and deterioration (if any) of the new ofes version versus the old one. They should be clear enough to readers without requiring to read the main sections. Statements about comparison of one single version versus observations are not enough. They can be additional information, but not enough as the main conclusions. Model users want to know whether they should use the new version or not, for what purpose it would be better to use the new version, for what purpose the old version is preferred?*
**Response:** We revised the abstract by adding "*This work suggests that OFES1 provides a reasonable multi-decadal estimate of global and basin-integrated warming trends above 700 m, except for the top 300 m for the Pacific Ocean and between 300–700 m for the Indian Ocean. Although the estimates of the global OHC during 1960–2016 are consistent with observations between 700–2000 m, caution is warranted while examining the basin-wide multi-decadal OHC variations using OFES1. The seemingly suboptimal OHC estimate based on OFES2, suggests that any conclusions on long-term climate variations derived from OFES2 might suffer from large drifts, necessitating audits*". (Lines#23–29).

Correspondingly, we added a third primary point in the conclusion section, as "*Overall, the global and basin-integrated OHC estimates for the period 1960–2016 were reasonable for the top 700 m upon considering the OFES1 results. Below 700 m, multi-decadal climate changes derived from the OFES1 need careful evaluations even though the estimates of global OHC between 700–2000 m are highly correlated with observations. The notable differences between OFES2 and EN4 suggest that attention is clearly warranted while concluding on multi-decadal climate changes based on OFES2.*" (Lines#938–942).

*3. I do see that English should be improved from place to place. Please use this chance to improve the English thoroughly.*
*Response:* This new revised manuscript was improved by Elsevier language editing services.